

# Shallow and Deep Convection Characteristics in the Greater Houston, Texas Area Using Cell Tracking Methodology

Kristofer S. Tuftedal[1], Bernat Puigdomènech Treserras[2], Mariko Oue[1], Pavlos Kollias[1,3]

[1]Division of Atmospheric Sciences, Stony Brook University, Stony Brook, NY, USA
[2]Department of Atmospheric and Oceanic Sciences, McGill University, Montréal, QC, Canada
[3]Center for Multiscale Applied Sensing, Brookhaven National Laboratory, Upton, NY, USA

*Correspondence to*: Kristofer S. Tuftedal *(kristofer.tuftedal@stonybrook.edu)*

**Abstract.** The convective lifecycle, from initiation to maturity and dissipation, is driven by a combination of kinematic, thermodynamic, microphysical, and radiative processes that are strongly coupled and variable in time and
space. Radars have been traditionally used to provide the convective clouds characteristics. Here, we analyzed climatological convective cell radar characteristics to obtain and assess the diurnal cycle of shallow, modest deep, and vigorous deep convective cells that formed in the Greater Houston area, using the National Weather Service radar from Houston, Texas and a multi-cell identification and tracking algorithm. The examined dataset spans over four years (2018–2021) and for the warm season months (June to September). The analysis showed the clear diurnal cycles
in cell initiation (CI), cell evolution parameters (e.g., maximum reflectivity, cloud top height, and the height of maximum reflectivity), consistent with the sea breeze circulation. The cell evolution is well represented by relationships between 1) the maximum radar reflectivity and its height, 2) the cloud top and the maximum vertically-integrated liquid, 3) the maximum reflectivity and columnar average reflectivity, and 4) cloud top ascent rate and cell lifetime. The relationships presented herein help to identify the cell lifecycle stages such as early shallow convection,
vigorous vertical development, anvil development, and convective core dissipation. We also analyzed the near-storm environment to address any differences in the environmental conditions present at the time of CI and how they may differ between convective type (shallow, modest deep, and vigorous deep cells).



## 1 Introduction

Convection is one of the most important contributors to the Earth's climate system through its transport of heat, moisture, and momentum. These processes strongly depend on cloud evolution. While an ordinal convective cloud model is proposed, the cloud evolution may vary depending on environment, diurnal cycle, etc. (e.g. Bony et al. 2015; Fridland et al. 2017; Ladino et al. 2017; Colin and Sherwood 2021). These variabilities may cause large uncertainties in convective parameterizations in numerical climate and cloud models. Modeling studies have attempted to answer some of these outstanding questions (Lee et al. 2008; Zhu et al. 2012; Varble et al 2014; Igel et al. 2015; Peters et al. 2019), but the general lack of high-quality observational data to compare with these modelling studies makes it difficult to assess the validity of the results presented therein. The ongoing debate in the literature about warm- and cold-phase convective invigoration (i.e., Sheffield et al. 2015; Fan et al. 2018; Abbott and Cronin 2021; Igel and van den Heever 2021; Grabowski and Morrison 2021) is one such example of the need for high-quality observational datasets to compare with modeling studies. The collection of a robust observational dataset that can be used as observational benchmark for modeling studies is challenging. Such a dataset requires a sufficient large sample size of convective clouds properties over a wide range of meteorological and aerosol conditions and convective cell centric methodologies rather than domain averaging approaches that fail to capture the lifecycle of convective clouds.

To address some of the observational shortcomings and needs, a diverse, interagency, coordinated effort took place in the greater Houston, Texas metropolitan area and surrounding region from the summer of 2021 through the summer of 2022 to collect a comprehensive dataset of isolated convective cells (Jensen et al. 2022).These efforts include the Tracking Aerosol Convection interactions Experiment (TRACER) supported by the United States Department of Energy (DOE) Atmospheric Radiation Measurement (ARM) facility and the National Science Foundation (NSF) Experiment of Sea Breeze Convection, Aerosols, Precipitation and Environment (ESCAPE) campaign. The study region was selected because it is warm and humid in the summer and commonly experiences onshore flow and sea breeze-forced convection. Recent studies on convective cells that form in coastal regions have illustrated that these cells are less influenced by the synoptic-scale meteorological conditions than cells that exist in only maritime or continental environments (e.g. Bergemann and Jakob 2016; Birch et al. 2016). The land-sea breeze circulation that develops in these regions have been shown to have a greater effect on local convection by acting as a forcing mechanism for convective initiation (CI) through an increase in surface convergence (Haurwitz 1947; Rotunno



1983). Study of convective cells that form under land-sea breeze circulation forced CI can be used to more directly

attribute convective characteristics to ongoing convective processes rather than the synoptic-scale meteorological

conditions present. This lack of dependence on the larger scale meteorology allows for more generalizable conclusions

to be drawn for any convective characteristics presented herein.

        Previous studies of long-term remote sensing data collected in Houston suggests that variability in convective

cloud microstructure, hydrometeor properties, and electrification is correlated to variability in aerosol conditions over

and downwind of Houston (Hu et al. 2019a, b). In these studies, NEXRAD radar observations were used to track

convective cells during different cloud condensation nuclei (CCN) conditions (satellite-retrieved; Rosenfeld et al.

2016), and to investigate how variability in cloud, precipitation and lightning characteristics related to CCN

conditions, though neglected proper control of meteorology. Also, a recent pilot study has identified the need to collect

observations of convective clouds at a temporal and spatial resolution higher than is possible from the existing

operational observations (Fridlind et al. 2019). Those studies show large variabilities of convective cloud properties

associated with cloud lifecycles. In addition, cloud lifecycles also vary from one individual cloud to another. These

substantial variabilities make it difficult to understand general characteristics of cloud lifecycle (evolution) by

analyzing the lifecycles of a few convective clouds. This lack of generalizable data also hinders our ability to evaluate

cloud model simulations.

        Here, we add and expand on these previous studies using a similar methodology, but focusing on both shallow

and deep convection and utilizing an extensive convective-cell centric methodology for the different types of isolated

convection observed aroundthe Houston metropolitan and surrounding areas. The climatologies of the observed

characteristics of shallow and deep convection are derived from the National Weather Service (NWS) KHGX Weather

Surveillance Radar–1988 Doppler (WSR-88D) and the Geostationary Operational Environmental Satellite-16 (GOES-

16). High Resolution Rapid Refresh model (HRRR; Smith et al. 2008; Dowell et al. 2022; James et al. 2022) data are

also used to elucidate differences in the NSE between convective cell types. Strict thresholding allows for the analysis

of the behaviors of each case type distinct from one another. Sensitivity testing is also performed to determine how

much the results presented herein vary with small variations in threshold selection. Section 2 outlines the data,

threshold variables, and analysis methods used, section 3 presents the analyses of these convective types, and section

4 discusses the observations and results.



## 2 Data and Methods

### 2.1 Domain and Data

The area used for this study was selected such that it was centered on Houston, Texas and extended ±125 km

to the north, east, south, and west, creating a 250 km by 250 km domain. This region was selected because the coastline

with the Gulf of Mexico generates regular land-sea breeze circulations which provide a forcing mechanism for CI.

This domain also provides regions of pristine and polluted aerosol regimes to the southwest and northeast respectively.

The area to the northeast of Houston is generally downstream of pollution sources, such as the Houston metropolitan

area itself and oil refineries near Houston allowing for the advection of this polluted air over this sub-region. The area

to the southwest of Houston is generally upstream and made up of mostly rural land even further upstream, allowing

the air here to be far less polluted.

Data from KHGX, GOES-16, and the HRRR were collected for the period of June through September 2018–

2021 where convective initiation occurred during local daytime (09:00–21:00 Central Daylight Time (CDT)).

Initiation during local daytime was chosen to ensure that sea breeze propagation was a primary forcing mechanism

for CI and to enable the analysis of GOES aerosol optical depth (AOD) data. Horizontal radar reflectivity factor ($Z_H$)

from KHGX was gridded to a 250 km by 250 km domain with 500 m by 500 m horizontal spacing each volume

coverage pattern (VCP), which was then used to estimate vertically integrated liquid (VIL), radar-derived cloud top

height (CTH), and radar-derived profile depth ($H_{cell}$). GOES-16 Channel 13 cloud top brightness temperature data

(GOESBT) were analyzed such that each 5-minute image was linked to the KHGX scan time nearest to each GOESBT

product time. GOES-16 AOD calculation requires cloud-free pixels to generate AOD data. In many cases, the location

of cell initiation already contains some form of cloud cover (be it low-, middle-, or upper-level clouds) at the time of

initiation, which inhibits AOD generation. When AOD values are generated, they are classified as low-, medium-, or

high-quality returns. To ensure a large enough sample size, AOD data denoted as medium- or high-quality were

temporally averaged for the 30 minutes preceding cell initiation at the location of cell initiation. HRRR data were

regridded to the same 500 m by 500 m grid used for $Z_H$. We calculated vertically integrated liquid (VIL) from gridded

$Z_H$ data from KHGX using the (1):



$$VIL\ (dB) = 10 * \log_{10}\left(\frac{\sum_{i=0}^{i=i_{max}} 3.44 * 10^{-6}[(Z_i + Z_{i+1})/2]^{4/7}dh}{1\ {kg}/{m^2}}\right)$$

*( 1 )*

where $Z$ is the radar reflectivity in units of $mm^6\ m^{-3}$.

**2.2 Cell Tracking**

KHGX VIL and GOES-16 observations were used as input to a modified version of the multi-cell identification and tracking (MCIT) algorithm (Hu et al. 2019a) to track all features with VIL of $\geq$ -20 dB during that period. The modifications made to the MCIT algorithm are provided in detail in Lamer et al. (2023). The MCIT

algorithm ingests time series of volume scans and tracks local maxima of VIL by identifying the two cells in consecutive radar scans that have common maximum VIL. In addition to the WSR-88D data, we used GOES-16 Channel 13 cloud top brightness temperature (GOESBT) to isolate cold topped and warm topped cells. The initial analysis from the modified MCIT algorithm identified 1,664,215 features with a VIL value $\geq$ -20 dB during the analysis period.

**2.3 Cell Classification**

To better characterize evolution of each cell, we employed the following cloud properties:

1)     CTH: the height at the middle of the highest gate with detectible signal from the WSR-88D;

2)     $H_{cell}$: the depth between the top of the highest gate of the radar detectible signal for a cell and the bottom of the lowest gate of the radar detectible signal.

3)     CRatio (Fig. 1): the ratio of the CTH to $H_{cell}$; and

4)     the tracked cell area based on VIL (Area)

Using these properties, GOESBT, and VIL, we classified tracked features into three categories: 1) shallow, 2) modest deep, and 3) vigorous deep convective cells. Thresholds for the classification are listed in table 1. These thresholds were empirically derived to avoid false classification of cloud systems such as high cirrus clouds, mesoscale

convective systems (MCSs), or large regions of stratiform precipitation. We also used the initial cluster fraction of the cell, which is the ration of the area of a given cell to the area of a cluster (parent) of the cell at the beginning of the cell lifetime, equal to 1 (a value of 1 meaning the cell is discrete). The split/merged cells are removed in this study.



The area threshold is used only to classify the shallow cells to ensure that these shallow features are not large regions of stratiform precipitation. This extensive thresholding removes noisy features, likely due to non-meteorological echoes (e.g., ground clutter, insects, etc.).


The shallow and deep convective cells selected based on these thresholds were then analyzed separately and compared with one another. Sensitivity studies were also performed on each case type by varying all thresholds (except initial cluster fraction and split/merge status) by ±5% individually and simultaneously to observe any changes in the distributions of certain variables for each case type (See Appendix).


**2.4 Climatological Analyses and Statistical Analysis**

Cell properties introduced in Sections 2.1 and 2.2 for all cells classified into the three categories during the four-year observation period were examined to allow for bulk analyses of cell type characteristics. Analyses used herein include observations during specific times during cells' lifetimes (such as the time of initiation), aggregates of all scans from the entire life of all cells of a given type, changes in variables over cell normalized lifetimes, diurnal changes in these variables, and spatial differences in initiation location. These analysis types enabled us to investigate how these cells changed as they grew, matured, and decayed, as well as allowed for the direct comparison of how these case types differ from each other.

To parse out any potential environmental controls on shallow and deep CI and intensity, HRRR model data and GOES-16 AOD data were analyzed. The HRRR data were mined to collect the convective available potential energy (CAPE), convective inhibition (CIN), various shear parameters, and vertical profiles of temperature, dew point, and wind speed and direction interpolated to the grid point of a given cells' initiation point for the forecast hour prior to each cells' initiation time. The AOD data were analyzed to investigate any role that aerosol loading may play in affecting CI. The nonparametric Mann-Whitney U test was used to investigate statistical differences between regional cell initiation AOD distributions (Mann and Whitney 1947). To further elucidate any differences between case types or within a given case type, days where the number of cells of a specific type exceeded the 95[th] percentile of the number of cells for days where those cell types were present were selected and analyzed separately.

**3 Results**



### 3.1 Overall Cell Characteristics

The monthly average number of cells varies little from month to month for June, July, August, and September. All convective cell types have the highest monthly average in August (Fig. 2). All three case types show little variability overall and no significant difference was found from month to month.

The storm motion could be important to understand storm evolution. To evaluate the primary speed and direction of cell motion, figure 3 shows frequency distributions of the propagation speed and direction as a function

of normalized lifetime for the three cell types investigated. The frequencies shown are normalized by the total number of samples at each normalized lifetime bin (every 0.025). All cell types tend to have storm motions primarily from the south to east. Shallow convective cells, overall, move the slowest out of the three cell types and have less variability in speed than modest and vigorous deep convective cell. Over the lifetime of these cells, shallow convective cell speed varies little and is much slower when compared to deep convection. Both deep convective case types tend to accelerate

with time with vigorous deep convective cells showing the most apparent tendency over their normalized lifetimes. Overall, for the three categories, the convective cells mainly tend to have storm motions spanning from southwest to east. Larger variability in storm motion is found at the later period of the cell lifetime for modest and vigorous deep cells, where the greatest variability is seen in the vigorous deep cells. This finding suggests that some of the vigorous deep convective cells may be supercellular in nature because of the large deviations from the early storm motions near

the time of initiation, but cannot be confirmed without further analyses of detailed conditions for supercells outside of the scope of this research. The direction of motion of sea-breeze induced convective cells can depend on the low-level (i.e., 925 hPa) wind direction relative to the coastline (Die Wang, 2023, personal communication). The direction of cell motion (particularly early in the lifetimes of these cells) indicates the possibility that the sea breeze along the Gulf Coast plays a part in storm initiation and propagation.

### 3.2 Location and Diurnal Frequency of Initiation

Figure 4 shows the diurnal frequency of initiation times as a function of local time. All cell types have their peak in initiation in the late morning/early afternoon hours, which then sharply decreases as the day progresses. Overall, all types of convection in this region preferentially initiate in the late morning/early afternoon. However, there is a slight difference in peak time. Shallow convection and vigorous deep convection show earlier peaks in

initiation time – between 10:00 and 14:00 CDT – when compared to the peak for modest deep convection, which is shifted slightly later (between 12:00 and 15:00; Fig. 4).

Figure 5 shows the initiation location for shallow (Fig. 5a), modest deep (Fig. 5b), and vigorous deep (Fig. 5c) convection. The three types all show a preference to initiate over land along the coastline to the southwest and northeast of Houston (within 100 km from the coast) with a local minimum in initiation over Galveston Bay,

suggesting that a land-sea contrast, hence sea breeze, is a key factor to initiate convection in this area. The inland propagation of the sea breeze can also be observed based on the cells' overall speed and direction of travel early in their lifetimes (Fig. 3). One feature of note is the obvious preference in shallow CI to the southwest of Houston (Fig. 5a). Another notable feature is that both moderate and vigorous deep convective cell types show a local maximum over the Houston metropolitan area, which is especially apparent in vigorous deep convective cases. This initiation

maximum could be caused by the enhancement from aerosol loading and/or urban heating, but will need further examination in the future.

To further investigate if the sea breeze plays any potential roll in cell initiation, we divided initiation locations shown in figure 5 into four 3-hourly bins to analyze the spatial and diurnal variability in cell initiation (Fig. 6). There is an obvious inland shift in initiation location, that is most obvious for vigorous deep convection, from the 09:00-

11:59 to 12:00-14:59 CDT period, further supporting the influence of the sea breeze on CI for all case types. The diurnal spatial shift in initiation location shows that the sea breeze is a key factor for CI along the coast. Figures 4 and 6 highlight that the early afternoon hours are the most preferable time for CI for all cell types. This is consistent with the previous studies (e.g., Park et al 2020).

The domain maximum in shallow CI to the southwest of Houston is also apparent in all periods except 18:00-

20:59 CDT (not shown), but modest and vigorous deep convective types do not visually show the same southwest initiation preference. Rather, the two deep convective types (especially vigorous deep convection) do show maximums in initiation over the Houston metropolitan area in the mid-to-late afternoon. We further speculate that this may be the result of urban heating allowing parcels to reach their convective temperatures during this time, even on days where CAPE is high, but CIN elsewhere in the domain is too strong to allow for other convection to initiate. However, as

stated previously, further research is necessary to elucidate what mechanism(s) is(are) responsible for this local initiation maximum in deep convection.



### 3.3 Diurnal Changes in Cell Characteristics

To assess diurnal cycle of convective cells, we analyzed cell hour-by-hour basis characteristics for each scan for the duration of its life over the course of the day. Figure 7 shows the diurnal trends in the distributions of GOESBT, the height of cell maximum radar reflectivity ($H_{dBZmax}$), and the cell maximum radar reflectivity ($dBZ_{max}$) for all scans of all convective cases. For shallow cells, the GOESBT tends to be constant at around 285 K over time, and the majority of the cells have $H_{dBZmax}$ below 5 km, suggesting warm precipitation processes. In contrast, the two deep convection types exhibit similar maximum frequencies of GOESBT before noon with a substantial shift of the maximum frequency to ~240 K for modest deep cells and ~220 K for the vigorous deep cells around noon, then taper off into the later evening and overnight hours. The frequency peak of $H_{dBZmax}$ for the modest and vigorous convective types is found below 6 km before noon, similar to shallow cells, which then shifts to 6–10 km for modest deep cells and 9–13 km for the vigorous deep cells until ~18:00 CDT for modest deep cells and 23:00 CDT for vigorous deep cells. The rapid change in GOESBT and $H_{dBZmax}$ shows the quick vertical evolution of these cells resulting in cold precipitation processes. CTH also shows the same change in characteristics as GOESBT and $H_{dBZmax}$ (not shown), further supporting late morning/early afternoon initiation. There is little dependency of maximum reflectivity on time.

### 3.4 Analysis of Bulk Cell Type and Normalized Cell Lifetime Characteristics

Even though the peak in CI has been shown to occur in the late morning/early afternoon, and peak in intensity in the early-to-mid afternoon, there may be diurnal variability and differences in duration in cell lifetime that are unaccounted for. To observe how these cells behave over the course of their respective lifetimes rather than the course of a given day, we have analyzed these cells by aggregating scans during specific periods of cell lifetimes, aggregating all scans of all cells of each type, and by normalizing each by its respective lifetime. These analyses allow for direct comparisons of case types, regardless of what time the cells initiated or how long they lasted.

The growth and decay of these cells can be seen by analyzing the change in the distributions of $dBZ_{max}$, $H_{dBZmax}$, maximum VIL, GOESBT, and the average of the maximum radar reflectivity for all columns within a given cell ($dBZ_{avg}$). Shallow convective cells show overall low $dBZ_{max}$ at low $H_{dBZmax}$ over the course of entire cell lifetimes (Fig. 8). Cell growth and decay is apparent, as $H_{dBZmax}$ and dBZmax shift to overall higher values during the first half of cell lifetimes and then decrease back to a distribution that looks most similar to cell initiation at the end of their lives. Modest and vigorous deep convection show clear signals of the birth (Fig. 8b,c), mature (Fig. 8e,f,h,i), and



dissipation (Fig. 8k,l) phases. Early in their lifetimes, these cell types are dominated by low $dBZ_{max}$ values at low

$H_{dBZmax}$, which both increase considerably moving into their mature phases. This trend is especially apparent in

vigorous deep convection, where early in cell lifetimes, there is a clear signal of initiation and some cells that have

begun to grow (based on the secondary maximum in high $dBZ_{max}$ at high $H_{dBZmax}$ values). As these cells continue to

mature, we see that $H_{dBZmax}$ remains high, but that $dBZ_{max}$ begins to decrease. This is indicative of cells where the

convective core has dissipated leaving "orphan anvils" (Hitschfeld 1960). However, $H_{dBZmax}$ appears to be slightly

skewed by bright banding, as both modest and vigorous deep convection show rather unnatural looking discontinuities

in their distributions around 6 km above radar level (ARL).

       Figure 9 supports that the feature seen in figure 8 around 6 km ARL to be caused by the bright band, since

there are no similar discontinuities in these distributions where GOESBT is warmer. As with figure 8, figure 9 clearly

shows the initiation, growth, and decay of these cell types. Shallow convective cells remain at relatively warm

GOESBT and low maximum VIL throughout their lives, whereas modest and vigorous deep convection show clear

growth from warm GOESBT, low maximum VIL cells to cold GOESBT high maximum VIL cells. Also shown as in

figure 8, is the dissipation phase, where GOESBT remains cold, but VIL drops off considerably. This further supports

the idea that this is the period when the convective cores have dissipated, leaving orphan anvils behind.

       To investigate potential anvil generation, the distributions of $dBZ_{avg}$ versus $dBZ_{max}$ are analyzed (Fig. 10).

When comparing shallow and deep convection, the feature of low $dBZ_{avg}$ and high $dBZ_{max}$ is indicative of anvil

generation. As with previous figures in this section, these anvil signatures are especially apparent in vigorous deep

convective cells. Early in deep convective lifetimes, cells have not had enough time to grow to a state where anvil

generation is possible (with the exception of the most intense cells). The vast majority of cells exist very near the one-

to-one line of $dBZ_{avg}$ to $dBZ_{max}$ with only a small subset of cells moving into the high $dBZ_{max}$ and lower $dBZ_{avg}$

indicative of cells growing and beginning to produce anvils (Fig. 10c). As cells move into the mature portion of their

lives, the low $dBZ_{avg}$/high $dBZ_{max}$ region, where strong, anvil generating cells are present, becomes the dominant

regime for these cells (Fig. 10f,i). As cells decay, the distribution maximum shifts back down near the one-to-one line

(Fig. 10l), which would be suggestive of orphan anvils, given the analyses provided in the previous paragraphs. One

feature of note in the vigorous deep convective cells during the dissipation phase is the secondary maximum in the

high $dBZ_{avg}$/high $dBZ_{max}$. We suspect that this is not a physical process and rather an artifact of cells leaving our

tracking domain prior to reaching dissipation.



The full lifetime distributions of $dBZ_{max}$, $H_{dBZmax}$, maximum VIL, GOESBT, and $dBZ_{avg}$ can be seen in figures 11 and 12. These plots show the full aggregation of scans considered in figures 8, 9, and 10 and further highlight how these cell types change over their lives. The majority of the shallow convective cells remain relatively warm-topped with low water contents, whereas deep convective cells begin in a similar parameter space as shallow convection and grow into cold-topped, high water content cells. The maximum in $H_{dBZmax}$ that we suspect is caused by bright banding is also clearly apparent. Based on Theil-Sen Estimator regressions performed on the GOESBT versus maximum CTH for all three case types, 6 km ARL would be approximately equivalent to a GOESBT of 260 K (Fig. 13). Theil-Sen Estimator regression was used to mitigate the effects on the maximum CTH distribution from convective cells where their maximum CTHs are artificially skewed downward when within the cone of silence for KHGX. While there is a secondary maximum in GOESBT at around 260 K for modest deep convection, the same maximum does not appear in vigorous deep convection, further suggesting that these maximums in $H_{dBZmax}$ are at least partially caused by bright banding.

Shallow convective cells reach their maximum CTH during approximately the first 10% of their life, appear to maintain this height until around 75%, and then gradually decrease in height until dissipation (Fig. 14a). Modest convective cells take substantially longer (~35-55% of the way through their life) to reach their maximum CTH and then sharply decrease in height during the last 15% (Fig. 14b). Vigorous convective cells, on the contrary, reach their maximum CTH quite quickly (within the first 15 to 35% of their life), maintain a tall cloud top, and then gradually decrease in height to dissipation (Fig. 14c). The gradual decrease in CTH leading to dissipation is further indicative of convective anvils. In this case, we believe this signal further supports that we are observing convective core dissipation, thus leaving orphan anvils behind. Both $H_{dBZmax}$ and $dBZ_{max}$ show similar trends to CTH for all case types. Shallow convective cells reach their $dBZ_{max}$ and $H_{dBZmax}$ early in the cells' normalized lifetime (within the first 20-30%), maintain, and then gradually decay to dissipation (Fig. 14d,g). Modest convection shows a gradual increase in $H_{dBZmax}$ that looks nearly identical to its maximum CTH. It appears that these cells reach their $H_{dBZmax}$ anywhere from 35 to 65% of the way through their lifetimes. However, $dBZ_{max}$ is reached much earlier (within the first 20% of their life) than $H_{dBZmax}$ (Fig. 14e,h). Vigorous deep convection reaches its $H_{dBZmax}$ much sooner in its normalized lifetime (between 10 and 25% of their normalized lives), like CTH (Fig. 14f). As with modest deep convection, vigorous deep convection also reaches its $dBZ_{max}$ early in its life (within the first 20% of their normalized life) and maintains these values until about halfway through their lives, when dissipation begins (Fig. 14i). One feature that is seen in $dBZ_{max}$



for both modest and vigorous deep convection is the apparent bimodality at the end of these cells' lifetimes, where it

appears some cells maintain high values of $dBZ_{max}$ and VIL (not shown) all the way up to dissipation (Fig. 14f,i). We

suspect that this is not a physical difference between some cells and others and is rather caused by some cells leaving

our observation domain while still in their mature stage instead of dissipating within the domain. This reasoning would

explain why $dBZ_{max}$ and VIL remain so high all the way up to the end of their life, as they are deemed "dissipated"

when leaving the domain, even though these cells may still persist for some time.

Cell echo base height ($H_{EBase}$) was determined using the maximum CTH minus the radar-derived echo profile

depth to estimate the precipitation base of these cells. For shallow convection, $H_{EBase}$ remains relatively flat for the

duration of these cells, but shows an increase in height during dissipation for modest and vigorous deep convective

cells (Fig. 14j,k,l). This signature is especially apparent in the vigorous deep convective cells. This increase in $H_{EBase}$,

in tandem with the minimal decrease in maximum CTH during dissipation of both types of deep convection (especially

vigorous deep convection), reinforces the idea that we are observing the generation of orphan anvils. The bimodality

here may also partially be caused by some cells leaving the domain during their mature phase, prior to dissipation, but

this is less certain than for the reasoning given for $dBZ_{max}$.

In an attempt to quantify vertical motion within these cells, we approximated this quantity three different

ways. We used the maximum CTH, and $H_{dBZmax}$ during each scan, and GOESBT at the time closest to radar scan time

with the time between samples to calculate maximum CTH and $H_{dBZmax}$ "ascent rates," as well as GOESBT "cloud

top lapse rates." As shown in figure 15, there are clear maximum CTH and $H_{dBZmax}$ ascent signatures early and descent

signatures late in the lives of these cells for all case types. GOESBT shows cooling during the same period where

ascent is seen and warming during the same period where descent is seen in maximum CTH and $H_{dBZmax}$. The near

identical timing and structure of these derived ascent rates suggest that these values may be good proxies for updraft

intensity during the early parts of these cell. However, the later portions of these cells' lives are dominated by the tops

or high upper portions of these features (as shown in figure 14) and most likely do not represent downdraft intensity

during dissipation. We suspect that, in vigorous deep convective cases, we may be seeing orphan anvils falling out as

virga since the descent signature occurs quite late in these cells' normalized lifetimes after their $H_{EBase}$ increases

considerably. Maximum CTH and $H_{dBZmax}$ descent rates and GOESBT warming rates match well late in these cells'

lives, which further supports the idea that we may be observing anvil fall-out. Further analysis is necessary to validate

these results. There is a hint of this signature as well for modest deep convective cells, but it is not as obvious. To

further assess the quality of these approximations, analyses outside the scope of this study will be needed.

**3.5 Near Storm Environment Analyses**

To elucidate any effects from the local meteorology where these cells formed, HRRR data for the hour

directly preceding cell initiation are extracted for the grid point where initiation occurs. The surface-based

environment does not appear to play a role in differentiating whether cells become shallow or deep convective cells.

All convective types initiated in environments where surface-based CAPE and CIN are similar overall. Storm relative

helicity (SRH) and bulk shear are also similar overall. Deep convective cases formed in environments with slightly

higher 0-1 km SRH, but the distributions of 0-1 km and 0-3 km SRH, as well as 0-6 km bulk shear are similar (not

shown).

When observing the composite soundings for the initiation locations for all convective case types, the

temperature profiles and parcel paths are also nearly identical. Dew point, however, is drier in the mid-to-upper levels

for the initiation locations of shallow convective cells when compared to deep convective cells (Fig. 16). While there

is substantial spread in the moisture profile in all case types, modest and vigorous deep convection moisture profiles

appear essentially the same. The wind profiles also appear relatively similar between case types. Surface winds are

slightly backed and veer with height to about 850 hPa, where they begin to back again (especially in shallow

convective cases.) However, the magnitude of these differences in mean wind are quite small (on the order of 5 kts

between 850 hPa and 600 hPa.) There are also apparent differences in upper-level winds, but again, are quite small in

magnitude.

In order to further parse out any differences in initiation environment, we select case days where the number

of cells that initiated on a given day, for a given case type, surpassed the 95th percentile of daily cell initiation (273,

61, and 43 cells for shallow, modest deep, and vigorous deep convective cells respectively). The composite soundings

for the initiation locations of the cell type that surpassed the 95th percentile of daily cell counts are shown in figure 17.

As with the soundings from figure 16, the key difference between these case types is that shallow convective initiation



locations tend to have drier mid-to-upper-level dew points. Near surface winds are nearly the same with some apparent differences in mid and upper-level winds.

Previously, we noted the discrepancy in initiation location of shallow convective cells that did not appear to be affecting deep convection. There is either some form of enhancement occurring to the southwest of the Houston metropolitan area or some form of suppression occurring to the northeast. One potential cause for this discrepancy could be the local meteorology in which these cells form. On days where shallow convective cells surpassed the 95th percentile of daily shallow convective cell initiations, we subdivided the domain to isolate the region to the southwest and northeast of Houston (Fig. 18a,b). The composite soundings for shallow convective cell initiation locations within these sub-domains are shown in figures 18c and 18d. While the moisture profile to the southwest of Houston is marginally drier in the mid-to-upper levels, these soundings are essentially identical. One interpretation of these results is that drier air aloft tends to inhibit convective updrafts from growing into deep convection. The entrainment of this drier air aloft would cause more evaporative cooling at the cloud top than a moister environment, leading to more negative buoyancy, and therefore limiting updraft intensity

To compare with modest and vigorous deep convection, we computed the total number of cells that formed within each sub-domain and for both sub-domains combined, then normalized them by the area of each domain. For shallow convection, the southwest domain had a cell initiation of 0.2074 cells km$^{-2}$, the northeast domain had a cell initiation of 0.1764 cells km$^{-2}$, and the combined southwest and northeast domain showed a total of 0.1920 cells km$^{-2}$. Modest deep convective cells had a cell initiation of 0.0389 cells km$^{-2}$ to the southwest, 0.0358 cells km$^{-2}$ to the northeast, and 0.0373 cells km$^{-2}$ combined. Vigorous deep convective cells had a cell initiation of 0.0331 cells km$^{-2}$ to the southwest, 0.0265 cells km$^{-2}$ to the northeast, and 0.0298 cells km$^{-2}$ combined. Overall, more cells initiated in the southwest area than the northeast area. These area-normalized initiation counts show that there may be some slight differences in deep CI between the areas to the southwest and northeast of Houston. These results combined with the near identical soundings for shallow convection suggest that something other than local meteorology is affecting shallow CI.

One potential ingredient that may influence CI and has been a key point of debate in recent literature is aerosol loading. We used temporal averages of GOES-16 AOD at the locations of cell initiation for the 30 minutes prior to MCIT detected initiation. We analyzed these data for the southwest and northeast sub-domains shown in Fig.



18a,b, but performed the analysis over the entire 4-year climatological dataset, rather than the subsets of days discussed

in the preceding paragraphs. The analyses of AOD for shallow and modest deep (Fig. 19a,b) cells at the locations of

their initiation show essentially identical distributions from southwest to northeast; the primary difference being that

both shallow and modest deep cells exhibit a longer tail toward higher AOD values to the southwest. Median (mean)

AOD for shallow convective cell initiation locations were 0.340 (0.400) to the southwest and 0.335 (0.366) to the

northeast, and 0.366 (0.448) to the southwest and 0.340 (0.379) to the northeast for modest deep convection.

Considering the substantial difference in initiation location for shallow convection and nearly identical initiation AOD

distributions, this result can be interpreted that bulk AOD does not play an important role in controlling cell initiation.

However, this does not mean that aerosol particles play no role as a control on cell initiation. Rather, it may be specific

species of aerosol particles that are more or less important to these processes. Vigorous deep convective cells,

however, do show differing distributions from southwest to northeast (Fig. 19c). As with shallow and modest deep

convection, vigorous deep convection also exhibits a longer tail extending to higher values of AOD to the southwest.

The initiation locations for vigorous deep convective cells tend to have marginally higher AOD values to the

southwest. The median (mean) of the AOD distributions for vigorous deep convective cell initiation locations was

0.433 (0.513) to the southwest and 0.366 (0.383) to the northeast. However, these differences in distributions are not

statistically significant. From the previous area-normalized cell initiation, we speculate that the marginally higher

values of AOD and rates of cell initiation to the southwest in vigorous deep convection suggest that aerosol loading

may indeed factor into vigorous deep CI, but that more marginal convective cells are either more dependent on specific

species of aerosol particles rather than overall aerosol loading or are not as affected by aerosol loading overall.

## 4 Summary and Discussion

The climatological characteristics of convective cell evolution and their diurnal cycles were analyzed using

the National Weather Service (NWS) Weather Surveillance Radar – 1988 Doppler (WSR-88D) from Houston, Texas

(KHGX) for the warm season months (June to September) from 2018 and 2021 and a modified version of the multi-

cell identification and tracking (MCIT) algorithm. In total, this study analyzed 52,216 convective cells (38,465 shallow

cells for 151,995 volume scans, 8,514 modest deep cells for 101,845 volume scans, and 5,237 vigorous deep cells for



105,666 volume scans). Analysis of these case types together allowed for the direct comparison of cell characteristics
        and the environments in which they form. The key findings from this study are:

1.  CI for all cell types occurs most frequently in the late morning/early afternoon over land, consistent with the
    inland incursion of the sea breeze front.

2.  There is a spatial variability in CI for shallow, modest, and vigorous deep convective cells, suggesting some
effects of aerosol loading and/or urban heating. This is particularly clear for shallow cells. Surface conditions
    do not appear to have any obvious effect on the resulting cell types that form in a given location. They exhibit
    slightly higher AOD values to the southwest of Houston, which is most easily observed when looking at the
    vigorous deep CI AOD distributions. While these results are not statistically significant, they suggest that
    aerosol loading may have some effect on deep CI. The initiation biases do not appear to be related to overall
aerosol loading based on the pre-CI AOD analysis with the exception of vigorous deep cells. Further analysis
    using high spatiotemporal aerosol and urban heat data will be needed.

3.  The CI location bias for shallow cells was coincident with slightly drier mid-to-upper-level moisture to the
    southwest of Houston, based on the HRRR reanalysis data. The deep convection categories do not appear to
    be related to the local meteorology at the point of initiation based on the HRRR reanalysis data. The surface-
based CAPE and CIN do not show significant relations with CI.

4.  The modest and vigorous deep convective cells particularly deepen in the afternoon/evening (12:00-21:00
    CDT) as the frequency peak of their heights of maximum reflectivity increase to 11 km and that of the
    brightness temperature decrease to 220 K. The shallow cells do not have clear diurnal variability in those
    parameters.

5.  The cell evolution is well represented by relationships between the following cell properties:

    1)  The maximum radar reflectivity and its height: The developing stage (cell lifetime normalized by cell
        duration < 0.75) is well represented by an exponential curve as the $H_{dBZmax}$ gradually increases from
        around 2 to 4 km with a maximum reflectivity of ~50 dBZ, which then dramatically increases to a height
        of 12 km for $dBZ_{max}$ values of 50-60 dBZ. These coincident increases in $dBZ_{max}$ and $H_{dBZmax}$ occur in
the early stage of cell lifetime (normalized lifetime < 0.5). The dissipation stage is represented by a wide
        distribution of the maximum reflectivity at a high altitude (~10 km) suggesting anvil development and
        convective core dissipation.



2) The brightness temperature (hence cloud top height) and the maximum VIL: The developing stage (normalized lifetime < 0.75) is well represented by an exponential curve as the brightness temperature gradually decreases from 290 to 260 K from the maximum VIL until 10 dB then dramatically decreases to 210 K for VIL > 10 dB.

3) The maximum reflectivity and columnar average reflectivity: As the cells begin to develop for all case types, these variables remain near the one-to-one line. However, as deep convective cells (especially vigorous deep convective cells) reach the middle phases of their lifecycles ($0.25 <$ normalized lifetime $\leq 0.75$), an obvious extension of high $dBZ_{max}$/low $dBZ_{avg}$ becomes apparent. The extension of the distribution during this period further supports the development of anvils. In the remaining 25% of vigorous deep convective cell lifetimes, the distribution of $dBZ_{max}$/$dBZ_{avg}$ shifts to low values for both, suggesting convective core dissipation leaving only orphan anvils behind.

6. The CTH ascent rate is slightly more positive (0.3 km/min) in the early stage (normalized lifetime < 0.4) and negative for the later stage (normalized lifetime > 0.8) for deep convective cells. Early in these cells' lives, the cell tops grow vertically as they intensify and the rates at which they ascent should be close to the actual updraft intensity. Late in the cells' lives, $H_{dBZmax}$ remains high aloft as it appears that $H_{dBZmax}$ remains in the anvil portion of the storm after it reaches maturity. These descent rates are likely to be representative of orphan anvils falling out as virga.

Based on the findings in this study, the analysis techniques presented can identify individual features within convective cells. Further parsing of these data may allow for the tracking of individual features within cells, such as tracking cores and anvils separately and being able to analyze their behaviors over their lives. Polarimetric variables can also be added for automated tracking of features such as differential reflectivity columns and arcs. The convective cells considered in this study only constitute about 3% of the features tracked during the climatology period and were selected based on the empirically derived thresholds in table 1. Some of the features we excluded by using these thresholds are non-meteorological in nature, but others include high clouds and large precipitation shields. Different empirically derived thresholds can be developed to isolate these and other features from the full dataset and used to create large climatologies of these features of interest. The analysis techniques presented in this study can also be

applied to the cloud resolving model simulations using radar simulator and cell tracking techniques (e.g. Oue et al.

2022). This will better evaluate the simulation results to understand isolated convective cell formation and evolution

mechanisms including effects of environmental factors such as aerosols.

**Appendix**

To assess the sensitivity of the results in this study, the thresholds were varied individually and

simultaneously by ±5% from the values presented in table 1. The number of cells selected in each sensitivity test were

recorded and plots were qualitatively analyzed to investigate substantial visual differences as these variables were

adjusted. Table A1 shows the number of cells selected for each variable adjusted. Figure A1 shows the visual

differences of the distributions based on changes in the most sensitive thresholds.

The number of shallow convective cases varied more for lifetime minimum GOESBT than any other variable.

The number of cases increased by 16.94% for a five percent reduction in the GOESBT threshold (from 268.0 K to

254.6 K) and decreased by 48.99% for a five percent increase in the same threshold (from 268.0 K to 281.4 K). The

visual differences in the distributions of cases are shown in figure A1a,b,c. As shown, the visual differences in the

distributions are caused by the addition or removal of colder topped shallow convection (Fig. A1c). The overall shape

of the distribution does not change otherwise. Increasing GOESBT only removes the more "intense" shallow

convective cells that have colder GOESBT values during their lifetimes.

          Like the shallow convective cases, the number of modest deep convection cases varied most with changes in

lifetime minimum GOESBT. There was a 27.01% reduction in cases with a five percent decrease in GOESBT (250.0

K to 237.5 K). This removes the slightly warmer GOESBT cloud tops, leaving an upper bound of more intense modest

deep convective cases (Fig. A1e). There was also a 23.94% increase in cases with a five percent increase in lifetime

minimum GOESBT (250.0 K to 262.5 K). This clearly shows the addition of warmer, lower intensity convective cells

(Fig. A1f). The modest convective cells also varied by more than 10% for the upper bound of the lifetime maximum

CTH, but there were no visible changes in distributions of these variable, only a reduction/increase in the number of

cases (Fig. A1g,h,i).

Vigorous convection varied most by lifetime maximum CTH. The number of cases increased by 28.36% for

a five percent decrease in maximum CTH (12 km to 11.4 km) and decreased by 26.79% for a five percent increase in

maximum CTH (12 km to 12.6 km). The variability in maximum CTH is shown in figure A1j,k,l and appears to behave similarly to how the sensitivity in maximum CTH affects modest deep convection. Adjusting this threshold, only appears to reduce or increase the number of cases and not change the shape of the distribution.


**Author Contribution**

Kristofer S. Tuftedal performed the threshold selection processes, all analyses presented herein, and the bulk of manuscript preparation. Bernat Puigdomènech Treserras wrote the script for the MCIT algorithm and generated the dataset analyzed in this paper. Mariko Oue provided assistance in the interpretation of results and assistance in preparing this manuscript. Pavlos Kollias provided assistance in the interpretation of results, suggestions for analysis techniques, and assistance as a Ph.D. advisor to Kristofer S. Tuftedal.

**Code Availability**

The MCIT tracking algorithm and analysis codes are available upon request.


**Data Availability**

The datasets generated for this study are available upon request. All other datasets used (WSR-88D KHGX, GOES-16, and HRRR) in this study are freely available through the National Centers for Environmental Information.

**Competing Interests**

The authors declare that they have no conflict of interest.

**Acknowledgements**



This study was supported by NSF Grant AGS-2019968 (Kollias and Oue) and the U.S. Department of Energy DE-

SC0021160 (Oue).

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





**Tables**

|  | Shallow Convection | Modest Deep Conv. | Vigorous Deep Conv. |
|---|---|---|---|
| Initial Cluster Fraction | 1 | 1 | 1 |
| Lifetime Min. GOESBT | ≥ 268 K | ≤ 250 K | ≤ 250 K |
| Lifetime Max. VIL | ≥ -10 dB | ≥ 0 dB | ≥ 0 dB |
| Lifetime Max. CTH | < 8 km | 8 ≤ CTH < 12 km | ≥ 12 km |
| Lifetime Max. CRatio | ≥ 0.60 | ≥ 0.75 | ≥ 0.75 |
| Lifetime Max. Area | ≤ 30 km$^2$ | N/A | N/A |
| Splits/Merges Removed | Yes | Yes | Yes |
|  |  |  |  |
| # of cells | 38465 | 8514 | 5237 |
| # of radar scans | 151995 | 101845 | 105666 |

Table 1: A table of the thresholds used to isolate shallow, modest deep, and vigorous deep convection from all tracked features from the modified version of the MCIT algorithm.




**Figures**

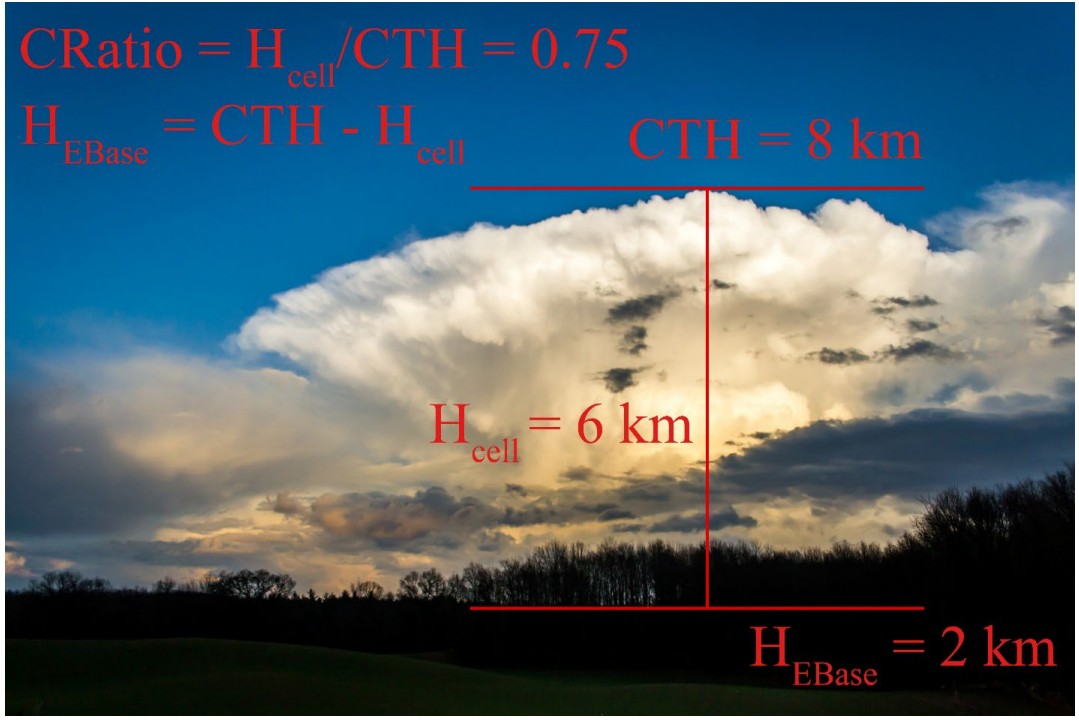

Figure 1: A visual illustration of CRatio, $H_{cell}$, and $H_{EBase}$. This image is meant to provide a visual context for these three variables. The actual values of CRatio and $H_{EBase}$ are calculated using the maximum CTH and $H_{cell}$, which are radar-derived quantities and will always be less than the actual height of the storm top and physical depth of a given cell.






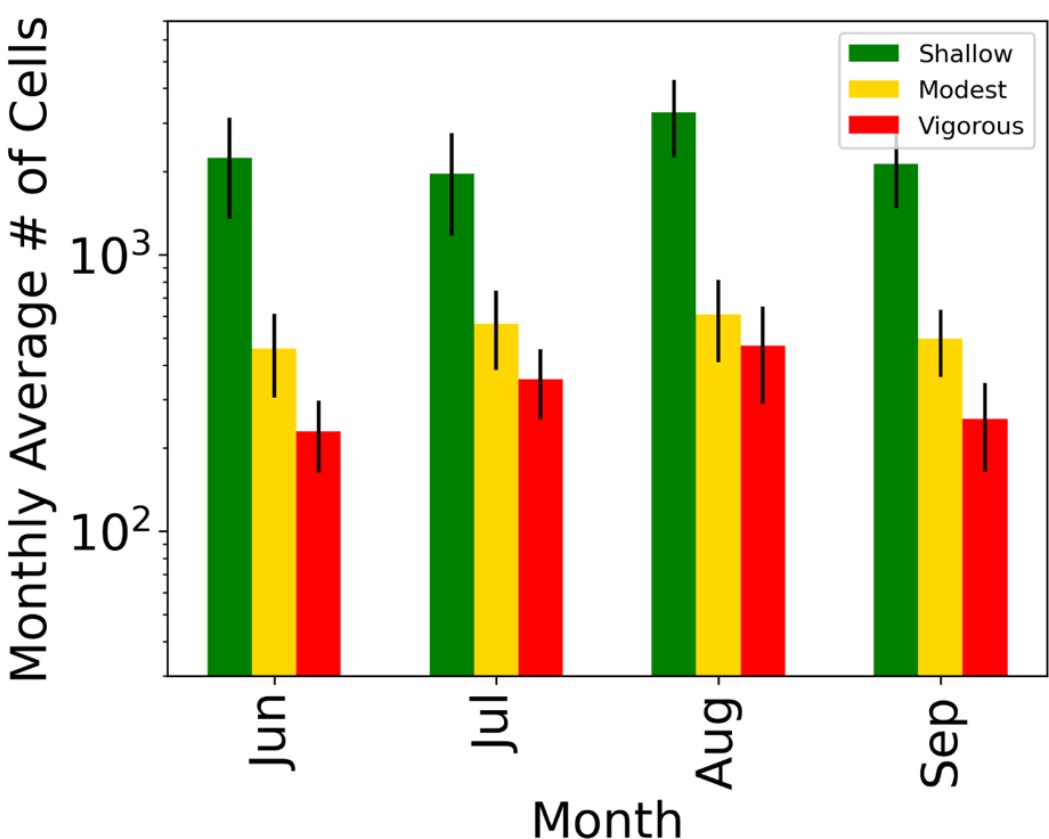

Figure 2: Bar graphs showing the monthly average cell count for shallow (green), modest deep (yellow), and vigorous deep (red) convection. The vertical black lines at the top of each bar denote ± one standard deviation.




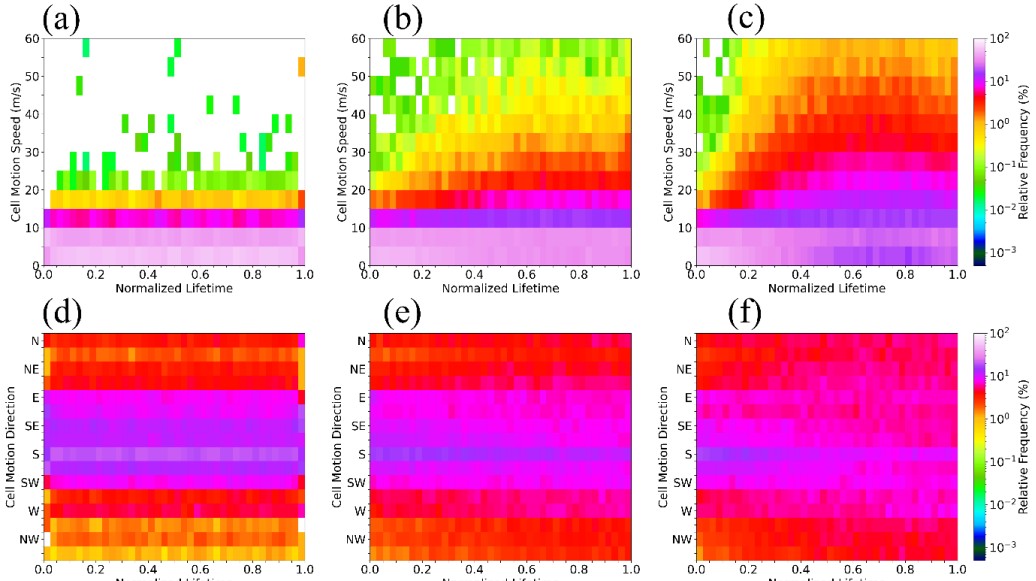

Figure 3: The normalized lifetime distributions of the bin count normalized cell motion speed (a, b, c) and cell motion direction (d, e, f) for all shallow (a, d), modest deep (b, e), and vigorous deep (c, f) convective cells.



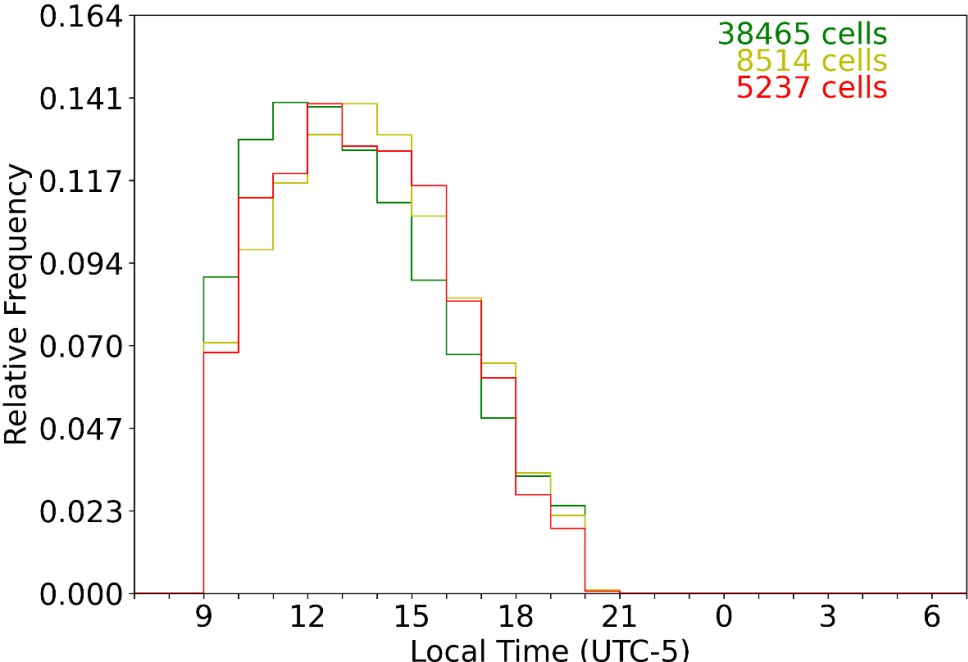

Figure 4: The frequency of initiation based on local time of day for (green) shallow convection, (yellow) modest deep convection, and (red) vigorous deep convection.



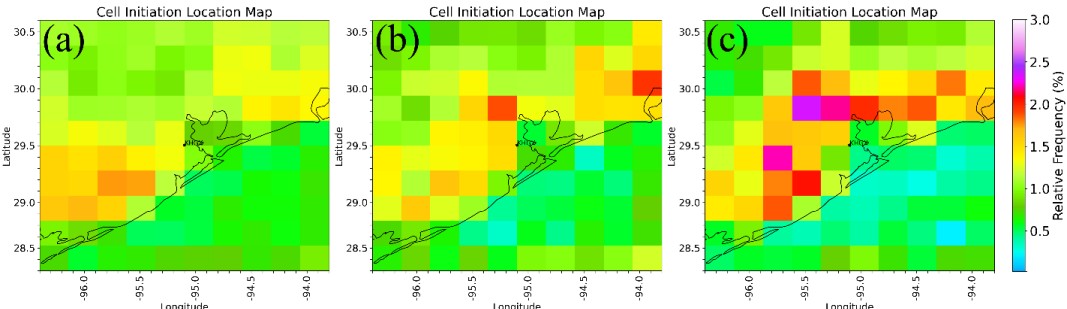

Figure 5: Maps showing the spatial distributions of initiation locations normalized by the total number of cells of each type for (a) shallow, (b) modest deep, and (c) vigorous deep convection. The black dot in each denotes the location of the KHGX WSR-88D radar used in this study.

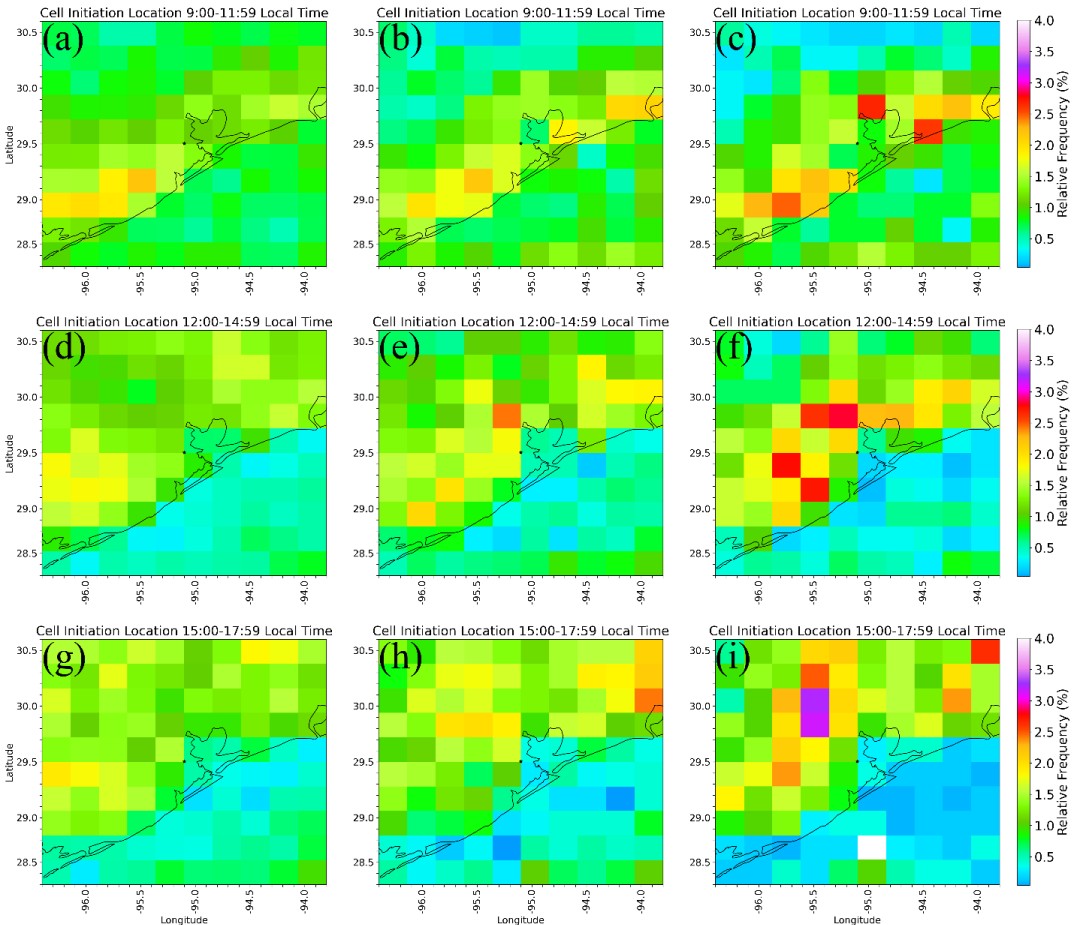

Figure 6: Maps showing the spatial distributions of initiation locations normalized by the number of cells that initiated during that period for the 3-hour periods 09:00 to 11:59 (a, b, c), 12:00 to 14:59 (d, e, f), and 15:00 to 17:59 (g, h, i), local time for shallow (a, d, g), modest deep (b, e, h), and vigorous deep (c, f, i) convection.

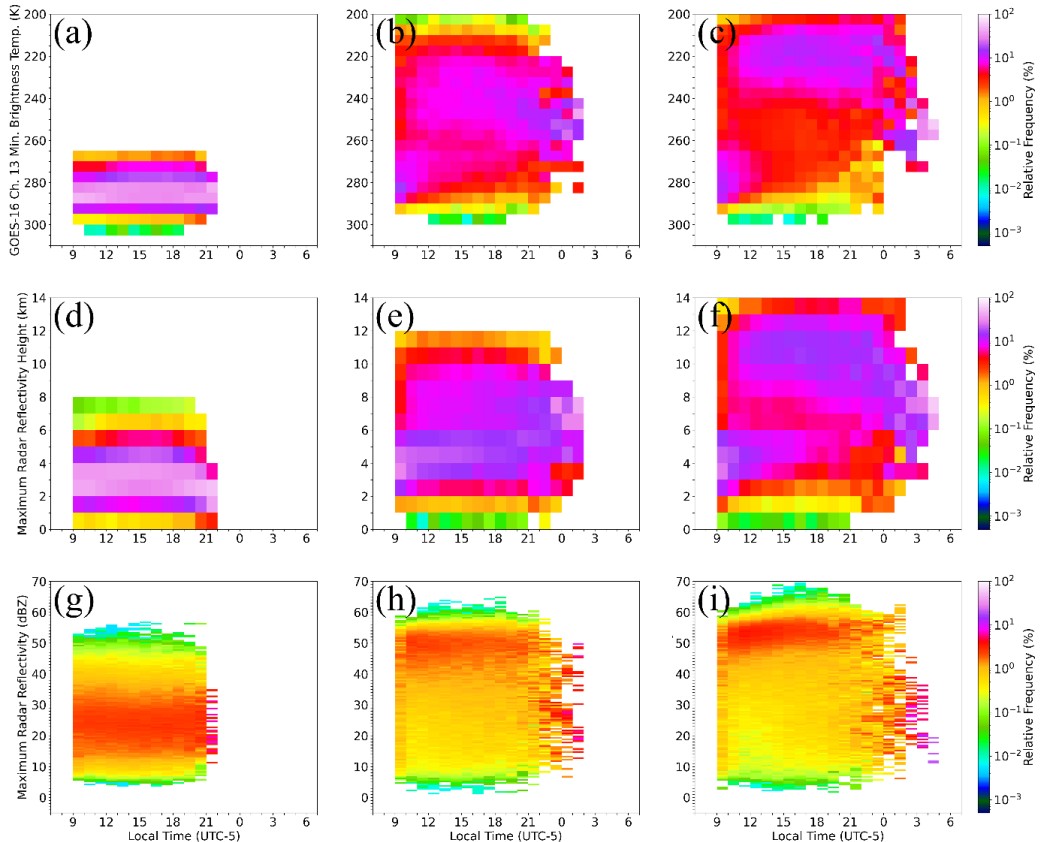

Figure 7: The time bin normalized distributions of GOESBT (a, b, c), $H_{dBZmax}$ (d, e, f), and $dBZ_{max}$(g, h, i) over the course of a day for all scans over the lifetimes of shallow (a, d, g), modest deep (b, e, h), and vigorous deep (c, f, i) convection.






Figure 8: The distributions normalized by the number of scans considered for each cell lifetime segment of $dBZ_{max}$ versus $H_{dBZmax}$ for the first 25% (a, b, c), the second 25% (b, e, f), the third 25% (g, h, i), and the final 25% of cell lifetimes (j, k, l) for shallow (a, d, g, j), modest deep (b, e, h, i), and vigorous deep (c, f, i, l) convection.







Figure 9: As in Fig. 8, but for maximum VIL versus GOESBT.



Figure 10: As in Fig. 8, but for dBZ$_{avg}$ versus dBZ$_{max}$. The dashed line shows the one-to-one value of dBZ$_{avg}$ and dBZ$_{max}$.

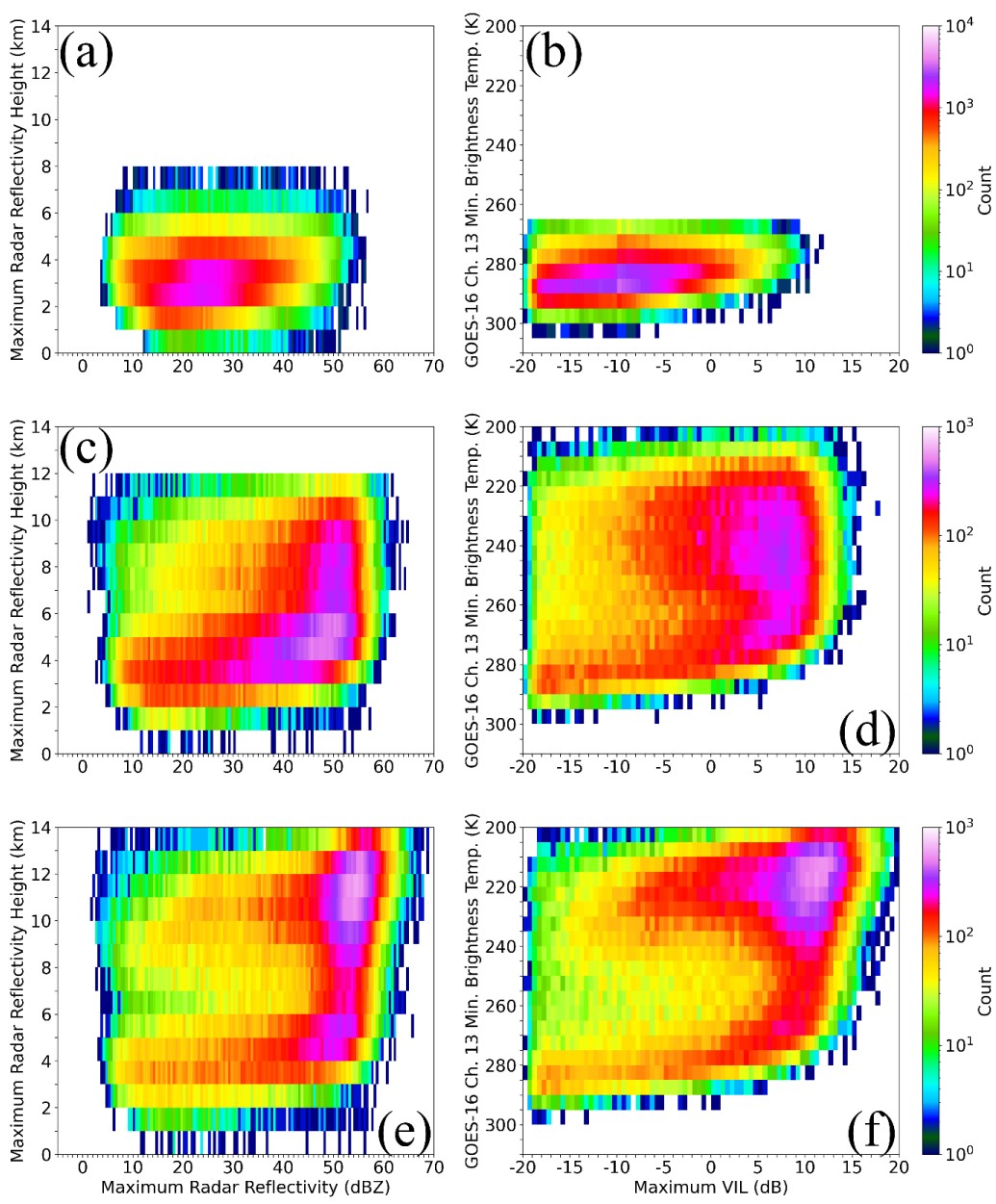

Figure 11: The aggregates of all radar scans for the entire lifetimes of all cell types for $dBZ_{max}$ versus $H_{dBZmax}$ (a, c, e) and maximum VIL versus GOESBT (b, d, f) for shallow (a, b), modest deep (c, d), and vigorous deep (e, f) convective cells.

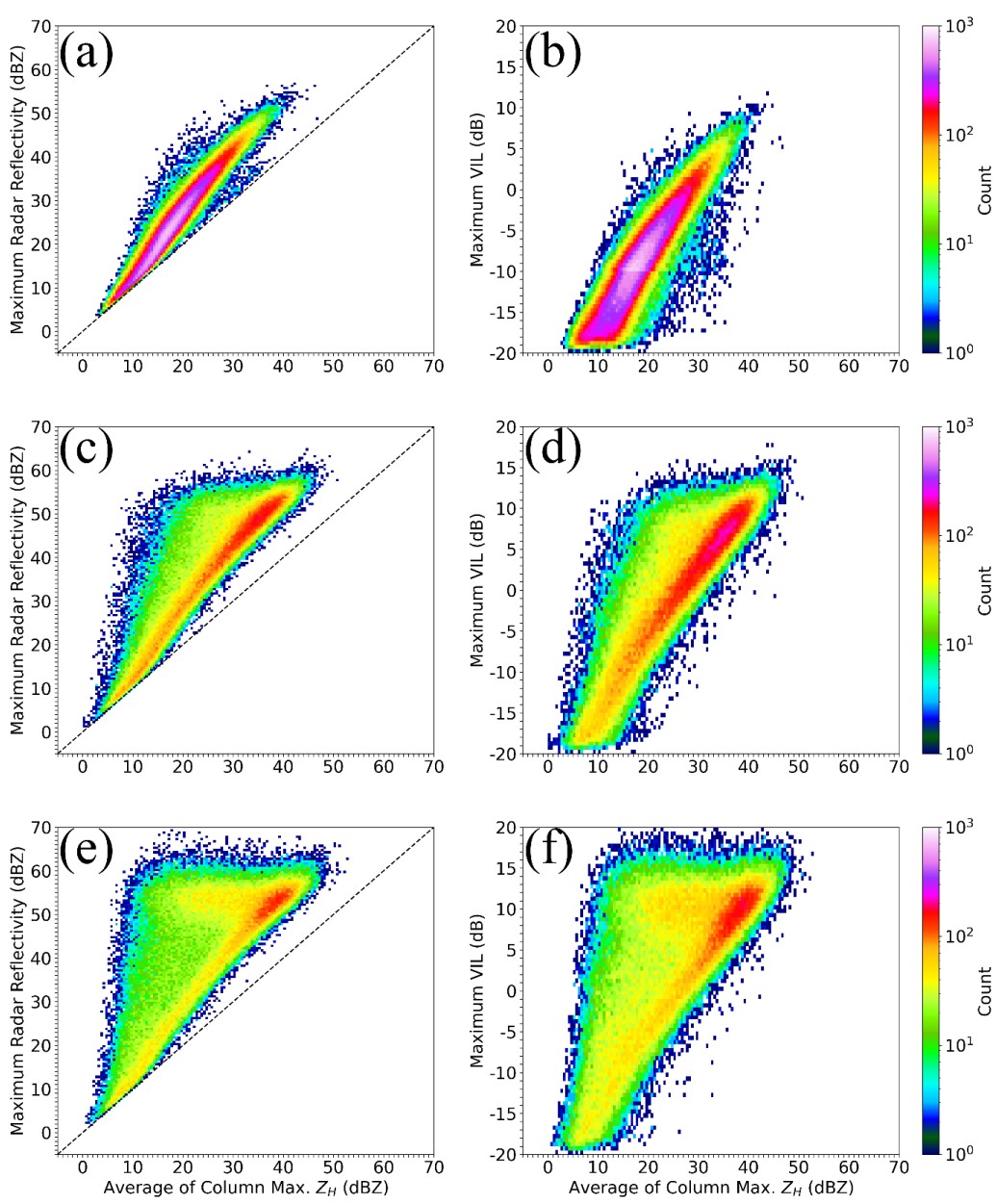

Figure 12: As in Fig. 11, but for the dBZ$_{avg}$ versus dBZ$_{max}$ (a, c, e) and dBZ$_{avg}$ versus maximum VIL (b, d, f). The
dashed line in (a, c, e) shows the one-to-one value of dBZ$_{avg}$ and dBZ$_{max}$.



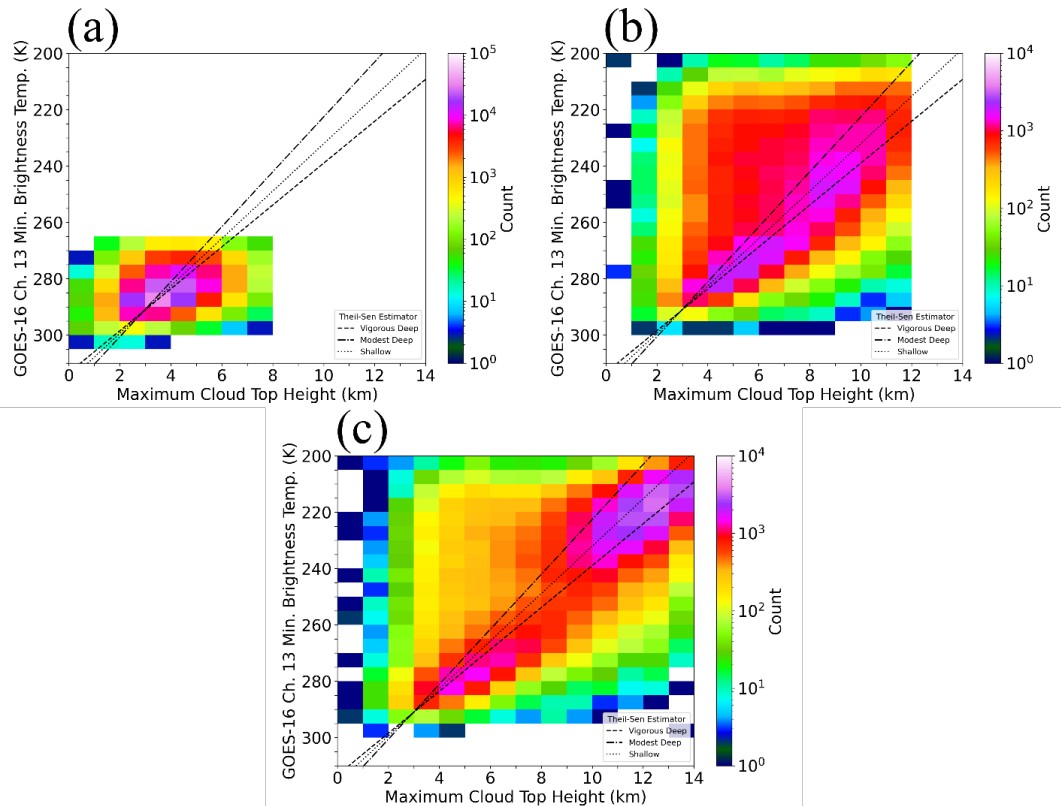

Figure 13: The distributions of CTH versus GOESBT for all scans of shallow (a), modest deep (b), and vigorous deep (c) convection. The lines denote the Theil-Sen estimator regression performed on shallow (dotted), modest deep (dot-dash), and vigorous deep (dashed) convection.



Figure 14: The bin normalized distributions for normalized lifetime evolutions of CTH (a, b, c), $H_{dBZmax}$ (d, e, f), $dBZ_{max}$ (g, h, i), and $H_{EBase}$ (j, k, l) for shallow (a, d, g, j), modest deep (b, e, h, k), and vigorous deep (c, f, i, l) convection.




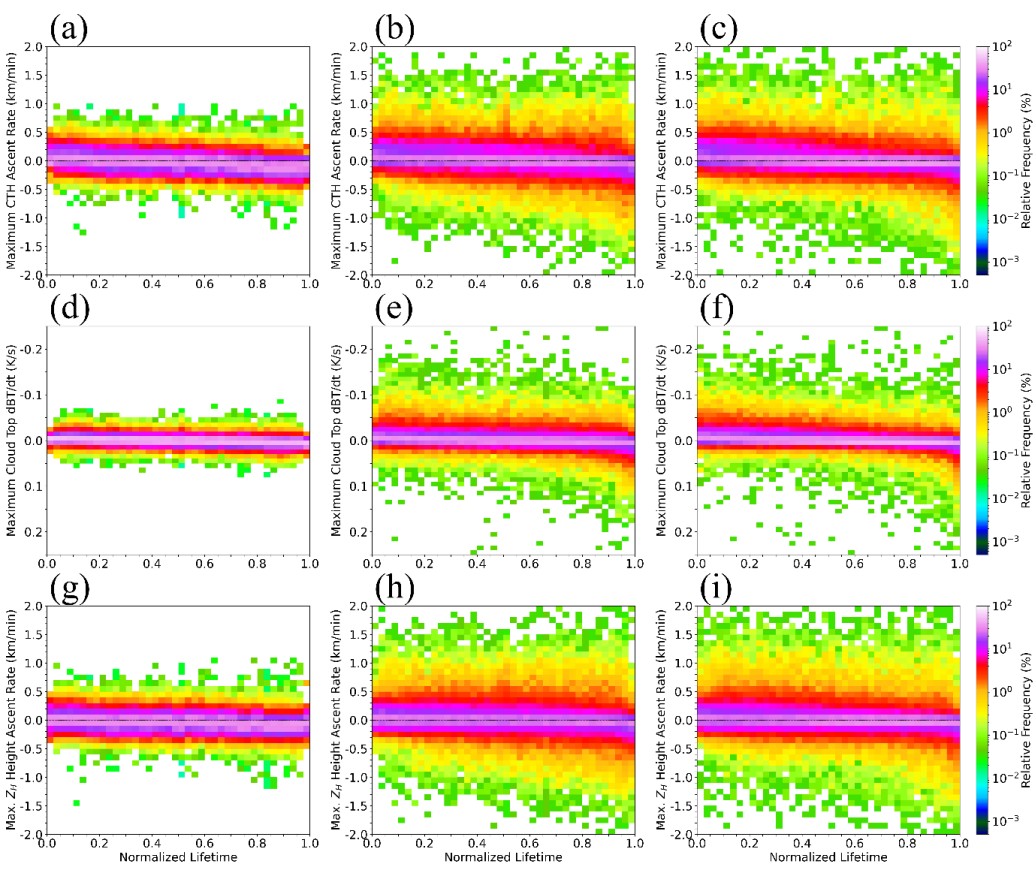

Figure 15: The bin normalized CTH (a, b, c), GOESBT (d, e, f), and $H_{dBZmax}$ (g, h, i) based ascent rates for shallow (a, d, g), modest deep (b, e, h), and vigorous deep (c, f, i) convection. The dashed line denotes the zero-change line.





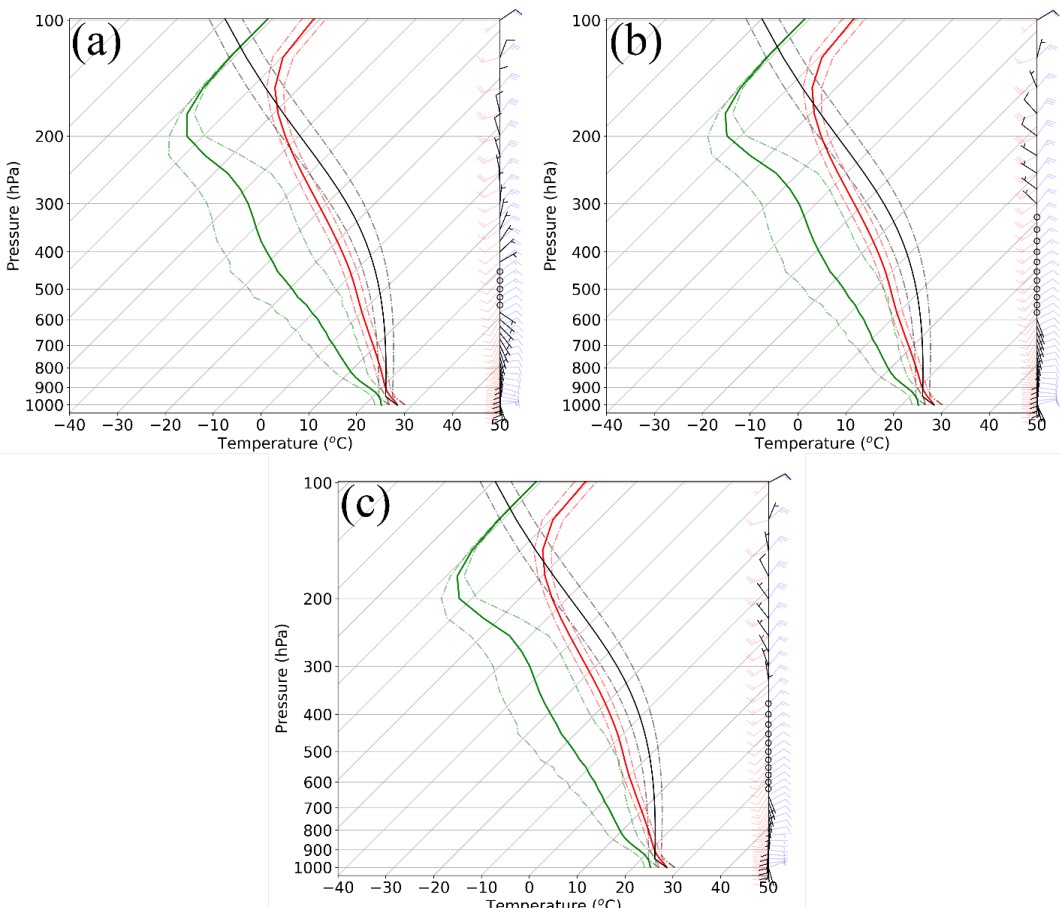

Figure 16: Composite HRRR soundings for the initiation location of shallow (a), modest deep (b), and vigorous deep (c) convection. The red, green, and black solid (dot-dashed) lines represent the mean (± one standard deviation) of temperature, dew point, and parcel path respectively. Black, red, and blue wind barbs represent the mean, plus one, and minus one standard deviation in knots.



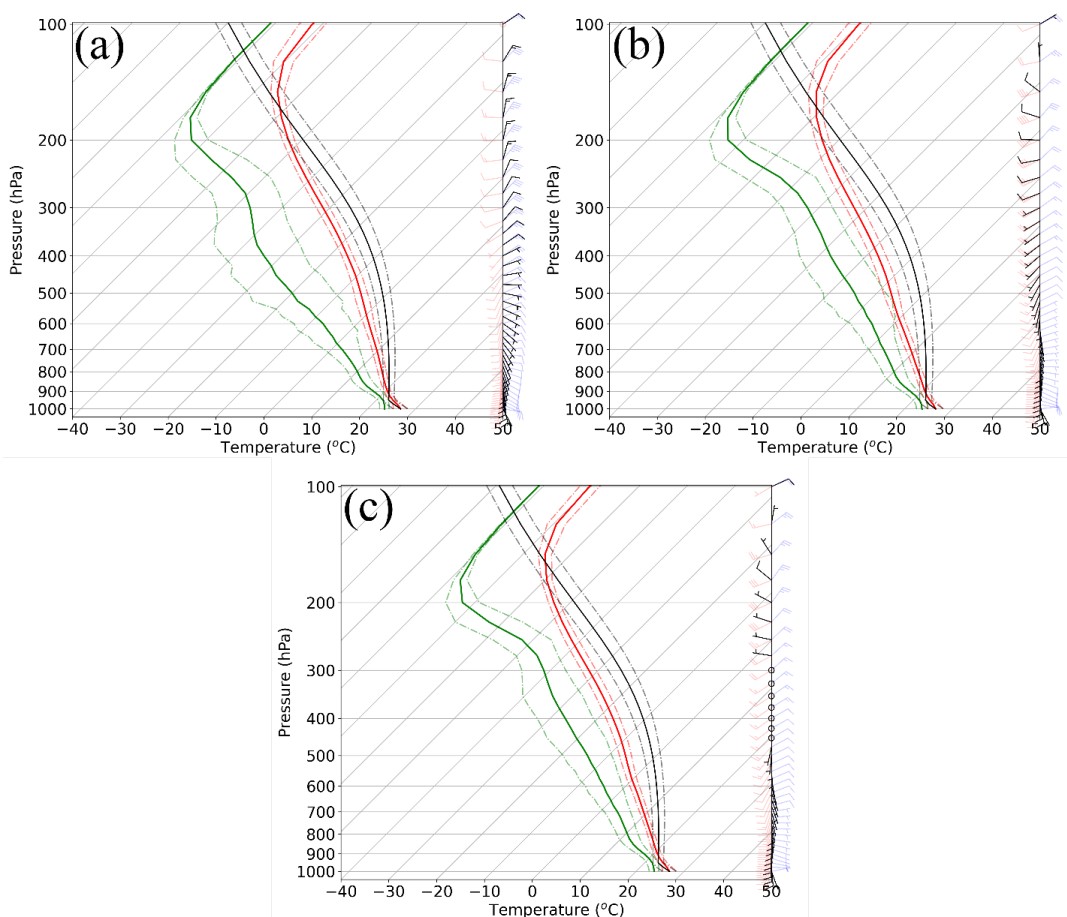

Figure 17: As in Fig. 16, but for days where the 95th percentile of daily cell count was surpassed for each case type.

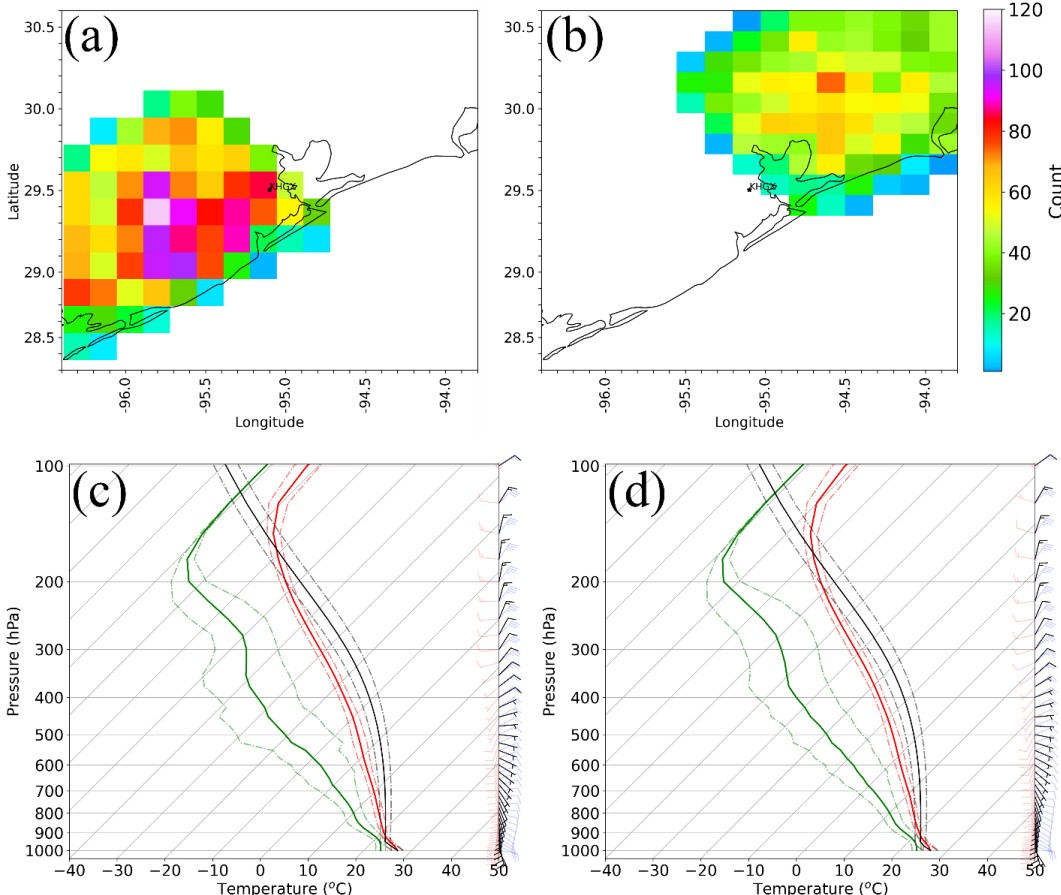

Figure 18: Subsets of the initiation locations for shallow convection on days where the 95[th] percentile of daily shallow
convective cell counts was surpassed (a, b) and the composite soundings for cells that initiated to the southwest (c)
and northeast (d) of Houston. Sounding plot depictions are as in Fig. 16.



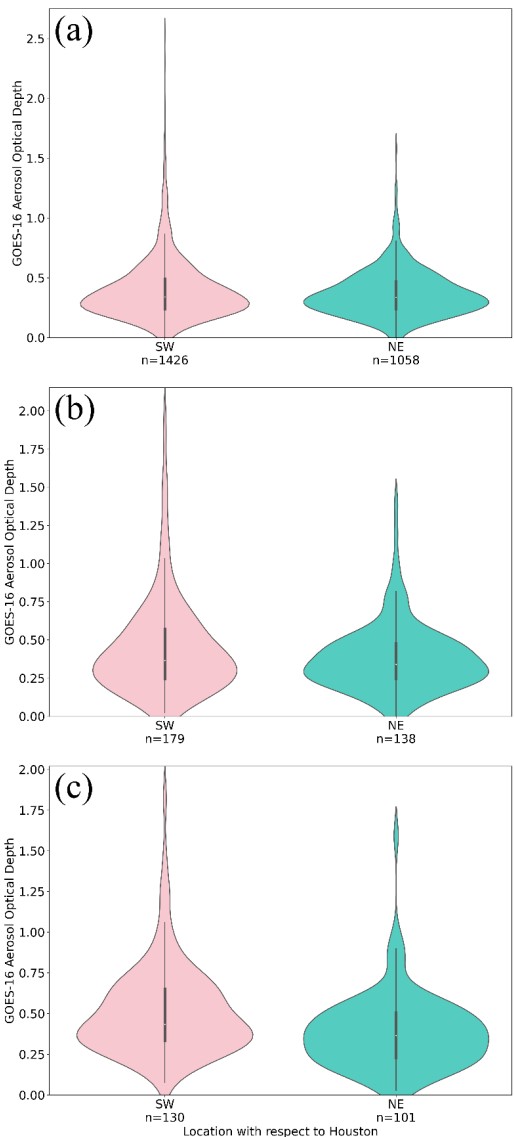

Figure 19: Violin plots depicting the distributions of the average of the 30-minute AOD values for the period prior to
cell initiation to the southwest (red; same region as Fig. 18a) and northeast (blue; same region as Fig. 18b) of
Houston for (a) shallow, (b) modest deep, and (c) vigorous deep convective cells. The number n under each violin
plot corresponds to the number of cells considered for each distribution.




**Appendix Tables and Figures**

| Shallow Convection | | | | |
|---|---|---|---|---|
| | -5% | | +5% | |
| | n | % change | n | % change |
| All Thresholds Simultaneously | 44560 | 15.85 | 19074 | -50.41 |
| Lifetime Max. Area | 37435 | -2.68 | 39363 | 2.33 |
| Lifetime Min. GOESBT | 44982 | 16.94 | 19620 | -48.99 |
| Lifetime Max. CRatio | 38518 | 0.14 | 38376 | -0.23 |
| Lifetime Max. CTH | 38285 | -0.47 | 38570 | 0.27 |
| Lifetime Max. VIL | 39389 | 2.40 | 37465 | -2.60 |

| Modest Deep Convection | | | | |
|---|---|---|---|---|
| | -5% | | +5% | |
| | n | % change | n | % change |
| All Thresholds Simultaneously | 5416 | -36.39 | 10843 | 27.35 |
| Lifetime Min. GOESBT | 6214 | -27.01 | 10552 | 23.94 |
| Lifetime Max. CRatio | 8535 | 0.25 | 8484 | -0.35 |
| Lifetime Max. CTH (Lower Bound) | 9214 | 8.22 | 7829 | -8.05 |
| Lifetime Max. CTH (Upper Bound) | 7029 | -17.44 | 9917 | 16.48 |
| Lifetime Max. VIL | 8570 | 0.66 | 8451 | -0.74 |

| Vigorous Deep Convection | | | | |
|---|---|---|---|---|
| | -5% | | +5% | |
| | n | % change | n | % change |
| All Thresholds Simultaneously | 6361 | 21.46 | 3879 | -25.93 |
| Lifetime Min. GOESBT | 5042 | -3.72 | 5349 | 2.14 |
| Lifetime Max. CRatio | 5241 | 0.08 | 5231 | -0.11 |
| Lifetime Max. CTH | 6722 | 28.36 | 3834 | -26.79 |
| Lifetime Max. VIL | 5245 | 0.15 | 5229 | -0.15 |

Table A1: A table showing the sensitivity of cell selection based on ±5% adjustments of a given variable. The number n represents the number of cells selected after the threshold adjustments were applied. Values for the percent change in the number of cells selected colored in green, yellow, and red denote cells with percent changes ≤ ±10%, ≤ ±20% and > ±20% respectively.

Figure A1: The lifetime distributions of GOESBT versus maximum VIL to illustrate threshold sensitivity for the most highly sensitive variables shown in table A1. The baseline (no change) distributions are shown in (a, d, g, j), 5% decrease is shown in (b, e, h, k) and 5% increase is shown in (c, f, i, l) for shallow convection GOESBT sensitivity (a,b,c), modest deep convection GOESBT sensitivity (d, e, f), vigorous deep convection CRatio sensitivity (g, h, i) and vigorous deep convection CTH sensitivity (j, k, l).