# Peer review of "Shallow and Deep Convection Characteristics in the Greater Houston, Texas Area Using Cell Tracking Methodology"

_EGUsphere, 2023_

## Author Response (AR1)

This study presents an extensive investigation into convective cell characteristics in the Greater Houston area, delivering important insights into their behavior, movement, anvil generation, vertical motions, and the influencing environmental factors. The paper is well-organized and distinguishes itself through its methodical tracking of convective cells across their lifecycle and the in-depth analysis of both shallow and deep convection phenomena. However, some aspects might require further refinement.

Specific comments:

1.  While the characteristics of convective cells and the impact of local meteorology are discussed, the exploration of the underlying causes and synoptic patterns of these meteorological conditions could be expanded. Such a discussion could provide greater context and depth to the study.

    **While we agree with this sentiment, the near-storm environment analyses have been removed from the paper at the request of reviewer #2.**

2.  The authors acknowledge potential artifacts and uncertainties in their data, primarily related to cells exiting the tracking domain during their anvil generation analysis. It will be beneficial to include a more comprehensive explanation of how this issue might influence the identified characteristics of shallow and deep convection.

    **Reviewer #2 suggested a similar adjustment to the paper. We have redone all analysis removing cells that have their first or last location as a domain border gridpoint. However, while these adjustments appear to have muted these artifacts slightly, they still remain. See our response to the Reviewer 2 comment #1.**

3.  The authors suggest that aerosols might not have a significant influence on cell initiation. However, it would be useful to understand the types of aerosols considered and the potential role each may play in convective processes. The current analysis and discussion on the role of aerosols could be made more explicit and comprehensive.

    **The data that we are using is raw aerosol optical depth from GOES-16, which does not distinguish between aerosol particle species. We have added speculatory discussion on what aerosols may be present in the region. It can be found on lines 337-340.**

4.  The study alludes to other potential factors influencing cell initiation but does not explore these in detail. Incorporating these factors into the analysis or outlining them as future areas of study could add more value and richness to the research.

    **We have added discussion of future areas of study with respect to cell initiation at**

**the end of the manuscript, lines 422-424.**

5. In Section 3.1, the authors state that all types of convective cells reach their peak monthly average in August. It would be beneficial if they could offer more insight into the underlying mechanisms driving these monthly variations. Understanding the causative factors may help further understand the influential factors for the convective cells.

   **We are unsure why this variability exists. We suspect this may be when the sea-breeze circulation is its most prevalent, but without further investigation, we cannot say definitively what causes this monthly variability.**

**Thank you for your comments that have helped aid in readability, further citations, and potential future areas of study for others to consider.**

Review of "Shallow and deep convection characteristics in the Greater Houston, Texas area using cell tracking methodology" by Kristofer S Tuftedal et al.

Summary of manuscript:

The authors describe the statistics of convective cells around Houston derived from 4 summer seasons of operational weather radar data. A tracking algorithm is used to capture cell life cycles and study initiation times and locations. Cells are classed as shallow, modest deep and vigorous deep based on their echo-top height and other characteristics, including GOES brightness temperature. Cells of all types are found to be most likely initiated in the morning, with a clear inland propagation associated with a sea breeze. Composite life cycles indicate distinct variations in timing of height of maximum reflectivity, and stages such as maturity and dissipation are identified. Storm initiation is related to near-storm environment based on coincident HRRR model data.

Review:

The manuscript is generally well-presented and contains useful results that are essential to other studies focusing on this domain (TRACER and ESCAPE campaigns) as well as methodology and results useful to the wider scientific community. A substantial number of clarifications are required to improve the understanding of the results, including the uncertainty. The manuscript feels slightly longer than necessary, and a number of figures are identified for potential removal without loss of key results. A section on the near-storm environment, while of great interest, does not seem to be supported by appropriate methodology or justification, and should be considered for removal. Overall, the manuscript should be considered for future publication after major revisions.

Major comments:

1. Incomplete cell life cycles.

There is some mystery about cells whose entire life cycle is affected by the domain. (i) Only in L260 (and again L292, L302) do the authors mention the possibility of cells leaving the domain. Do these cells still contribute to the statistics? How does that impact the life cycle statistics? Or the cell classification (e.g. a cell that was shallow in the domain but may have grown to be deep)? Would it not be prudent to remove all cells that leave the domain? Given the large number of cells, presumably this would leave a sufficient number for most statistical results. As a benefit, it would provide cleaner result and remove at least three artefacts (as suggested in L260 and L292, L302) and give a clearer picture of the dissipating stage. (ii) Only in L270 do the authors mention the radar "cone of silence" and some mitigation. What, exactly, happens to these cells? Are their statistics still incorporated? Is only their CTH corrected, or also maximum dBZ, etc?

**We appreciate the reviewer's comment. Taking into count this comment, all analyses have been redone to remove cells who have a first or last position recorded on the domain edge where the domain edge is defined as the first or last grid point on the edges of the**

domain. Removing these cells reduced our sample size by 2,469 shallow cases (from the initial 38,465 to 35,996), 579 modest deep cases (from 8,514 to 7,935), and 368 vigorous deep cases (from 5,237 to 4,869).

Oue et al. (2022, their Fig.3) suggested that the radar "cone of silence" attributed to the radar scan strategy could produce underestimation of VIL within 10 – 20 km distance from the radar. To mitigate the effects of the radar cone of silence, we remove all cases that pass within 15 km of KHGX. Domain departure and "cone of silence" mitigation steps are now outlined in the methodology.

Below I have included the original plots and revised plots (which remove cells that begin or end their life at the edge of the domain). Interestingly, there are still double peaks in the frequencies of $dBZ_{max}$ (Fig. b,c,e,f) at the later stage of the cell life (normalized lifetime >0.4). This is particularly apparent in the modest cells. This suggests that some of the modest cells that had the lower ETH were dominated by warm phase similar to the shallow cells, as the maximum reflectivity was found below 6 km through the lifetime. We replaced the Fig.10 with hew new figure and added the discussion to the revised manuscript.

Oue, M., S. M. Saleeby, P. J. Marinescu, P. Kollias, and S. C. van den Heever, 2022: Optimizing radar scan strategies for tracking isolated deep convection using observing system simulation experiments. Atmos. Meas. Tech., 15, 4931–4950, doi:10.5194/amt-15-4931-2022.

Original Figure.

[Figure]

**Revised Figure applying domain entrance/exit mitigation.**

[Figure]

2. Height uncertainty.

A second concern is the uncertainty in height, which is a major variable of interest for this study. There will be uncertainty due to the methodology, which is to pick the centre of the beam where either a detectable signal is found (CTH) or where the highest reflectivity is found (Hdbzmax). This brings both uncertainty due to the beam width and due to the separation between scans for the VCP. Both these uncertainties will increase with range from the radar and will be substantial (>1km) towards the edge of the domain. Hence, the paper would benefit from a consideration of uncertainty due to the VCP. In particular, results such as those in Figure 15 could be considered in context of this uncertainty.

**We agree with the reviewer. The NEXRAD VCPs could produce uncertainties in estimating CTH (ETH in the revised manuscript). We have added discussion about this height uncertainty and how it may affect the results presented therein from line 330 to 334 in the new version of the manuscript. Despite the uncertainty in the ETH estimates, the tendency of the ascending rates as a function of normalized lifetime are well consistent with those from GOESBT.**

3. Near-storm environment and HRRR.

The third main concern regards the near-storm environment and the use of HRRR to analyse this. Firstly, very little information is provided on HRRR in this study (see minor comments as well). Most importantly, the authors should provide evidence that HRRR is an appropriate tool to capture near-storm environment, particularly CAPE, CIN, and storm-relative helicity. Does the HRRR capture the sharp vertical gradients associated with CIN and dry layers? Won't such gradients be smoothed out during the data assimilation cycles and/or due to the model's effective resolution? Does HRRR really capture km-scale variability that the authors rely on for their composite soundings? Is HRRR good enough at predicting the convective storm populations that its model soundings can reasonably be used to relate to observed convective storm properties?

Regarding composite soundings, it is not obvious that this is the appropriate tool to study differences in environment. Any specific features of interest (such as CIN or dry layers) will be smoothed out by compositing. If anything, it would be more helpful to provide PDFs of CAPE and CIN across all the soundings (provided the HRRR soundings can be justified). As it stands, the authors may wish to consider removing this section 3.5, as the paper contains sufficient results of interest without it. Alternatively, the authors could consider discussing 1-3 case studies with significantly different environments, and suggest from that discussion a way forward to do further composite analysis.

**We have decided to remove the NSE based on HRRR, but continue to include the GOES-16 aerosol optical depth analysis since these are based on observations rather than model analysis and is more in-line with the scope of the study.**

Minor comments:

52: "lack of dependence on the larger scale meteorology" – There is, of course, a mesoscale phenomenon (the sea breeze) as an underlying forcing mechanism of the convection in this study (L47-49). To what extent are the findings from this study really generalisable to other synoptic situations?

**We are observing processes within convection that are independent of synoptic scale forcing to assess the processes inherent to all convection. Synoptic scale influences can greatly affect storm type. While we have not removed shallow or deep convection triggered by more synoptically driven events, we have shown that the vast majority of isolated convective precipitation cells are triggered by sea-breeze propagation. Since this sea-breeze initiated convection does not rely on synoptic scale support, the processes therein should be indicative of convective processes within all convection which can then be further modified by synoptic influences.**

72: Please expand (and possibly define) NSE on first use.

**Removed this because of the removal of the near-storm environment analysis.**

72-73: "Strict thresholding allows for the analysis of the behaviors of each case type distinct from one another." – Please expand on (a) what variables you will threshold and (b) what is meant by "each case type". Is case type shallow or deep, or something rather different? If you threshold on lots of variables, give a couple of examples here.

**These are the variables discussed in the methodology section. We added a section to better clarify what is being described in lines 74-76 in the new version of this manuscript.**

83-87: Are there any references or specific values the authors can provide here for the level of pollution? There is a difference between "pristine" and "far less polluted". Presumably, other TRACER or ESCAPE studies have reported on aerosol concentrations?

**The authors are unaware of studies from TRACER/ESCAPE or other observational studies that report "typical" (or any) aerosol concentrations for these regions. The naming convention for these regions is relative to one another based on the land use of each area and the general flow pattern over the domain. We have added text in lines 85-90 in the new manuscript to better clarify this passage.**

88: "the HRRR data" – Please provide a separate (sub-)section that briefly describes the HRRR data. Particularly, for the 0900-2100 CDT period, what are the initiation times? For example, given the hourly refresh, are you always using data valid at T+3hrs, or are you using data from a fixed initiation time? What is the grid resolution? What output will be used?

**This has been removed because of the removal of the near-storm environment analyses.**

90: Does daytime initiation "ensure" that sea breeze propagation was the mechanism or does it "increase the likelihood"?

**Good point. This sentence has been adjusted to reflect this point and is at lines 92-94 in the new manuscript.**

88-95: Please provide a bit more information on the KHGX. How long does one VCP take? What is the beam width? Consequently, what is the horizontal width at 125km range? It is important for the reader to understand the original resolution of the data rather than the 500m by 500m horizontal grid spacing used for the regridded data.

**KHGX is a typical National Weather Service WSR-88D radar (Crum 1993; Radar Operations Center 2022). These radars operate under several different VCPs which have changed over the years (Radar Operations Center 2015; Zittel 2019). Even within these VCPs, there are dynamic scanning techniques such as the Automated Volume Scan Evaluation and Termination, Supplemental Adaptive Intra-volume Low-Level Scanning, and others that forecasters can use (Chrisman 2009; 2013; 2014; 2016). These all have their own effect on how long it takes an individual VCP to complete. That said, it typically takes on the order of about 5 minutes to complete one VCP when the radar is operating in precipitation mode. The MCIT tracking algorithm does not track what VCP or additional scanning technique that the WSR-88D is utilizing. As far as the beamwidth, the native beamwidth of all WSR-88D radars is 0.925º (Radar Operations Center 2022) which is oversampled to 0.5º, providing a ~1.1 km horizontal width at 125 km range. The information/citations provided here have also been added at lines 95-102 to provide clarity to the reader.**

**Chrisman, J. N., 2009: Automated Volume Scan Evaluation and Termination (AVSET)—A simple technique to achieve faster volume scan updates. 34th Conf. on Radar Meteorology, Williamsburg, VA, Amer. Meteor. Soc., P4.4, https://ams.confex.com/ams/34Radar/techprogram/paper_155324.htm.**

Chrisman, J. N., 2013: Dynamic scanning. NEXRAD Now, 22, 1–3, https://www.roc.noaa.gov/WSR88D/PublicDocs/NNOW/NNow22c.pdf.

Chrisman, J. N., 2014: Multiple elevation scan option for SAILS (MESO-SAILS)—The next step in dynamic scanning for the WSR-88D. Radar Operations Center, 27 pp., https://www.roc.noaa.gov/wsr88d/PublicDocs/NewTechnology/MESO-SAILS_Description_Briefing_Jan_2014.pdf.

Chrisman, J. N., 2016: Mid-volume rescan of low-level elevations (MRLE): A new approach to enhance sampling of quasi-linear convective systems (QLCSs). New Radar Technologies Web Page, NOAA/NWS/Radar Operations Center, 21 pp., https://www.roc.noaa.gov/WSR88D/PublicDocs/NewTechnology/DQ_QLCS_MRLE_June_2016.pdf.

Crum, T. D. and R. L. Alberty, 1993: The WSR-88D and the WSR-88D operational support facility. *Bull. Amer. Meteor. Soc.*, 74, 1669–1688, doi: 10.1175/1520-0477(1993)074<1669:TWATWO>2.0.CO;2.

Radar Operations Center, 2015: WSR-88D Volume Coverage Pattern (VCP) improvement initiatives. New Radar Technologies Web Page, NOAA/NWS/Radar Operations Center, 8pp., https://www.roc.noaa.gov/WSR88D/PublicDocs/NewTechnology/New_VCP_Paradigm_Public_Oct_2015.pdf.

Radar Operations Center, 2022: NEXRAD technical information. NEXRAD Technical Information Web Page, NOAA/NWS/Radar Operations Center, https://www.roc.noaa.gov/WSR88D/Engineering/NEXRADTechInfo.aspx.

Zittel, W. D., 2019: Theory and concept of operations for multi-PRF dealiasing algorithm's VCP 112. New Radar Technologies Web Page, NOAA/NWS/Radar Operations Center, 13 pp., https://www.roc.noaa.gov/WSR88D/PublicDocs/NewTechnology/Theory_ConOps_VCP112.pdf.

104: Is this a standard equation for VIL? If so, please provide a reference. In any case, please provide interpretation of the formula and justification. Should the summation go to imax or imax-1 (and what is imax)? Is the denominator relevant if it is set to 1? Presumably, dh is the vertical spacing between individual sweeps of the VCP. As such, the separation will increase with range and the VIL will be quite uncertain at greater distance. Also, the highest sweep may not capture cloud tops close to the radar. Please mention these limitations and uncertainties in the text and possibly even quantify the uncertainty. Perhaps it is best to incorporate this comment with my previous one, and provide an expanded overview of the KHGX data and the subsequent derivation of VIL.

Later on, values for VIL > -20 dB are mentioned. It would be helpful for the reader to have an understanding of typical values (perhaps referring to the literature and/or more widely known variables such as total water path) for shallow, congestus, and deep convection.

**Yes, this is the standard equation for VIL (Greene and Clark 1972; Amburn and Wolf 1997). However, we are reporting it in units of dB rather than the standard unit of kg m$^{-2}$ for ease of interpretation. The denominator is there simply for completeness for unit cancelation and proper calculation of VIL in dB units. We are using a summation from 0 to imax-1 and have corrected the equation in the text. We have also referenced Oue et al. (2022) to address VIL uncertainty with distance. As far as "typical" values go. We have been unable to find literature that contains such values. The closest we could find was Amburn and Wolf 1997 and a subsequent unpublished study by the National Weather Service office in Nashville, TN (National Weather Service Web Page), which uses VIL and VIL Density to estimate hail size. They provide values for vigorous convection which range from 0.25 kg m$^{-2}$ (-6.02 dB) on the low end to 72 kg m$^{-2}$ (18.57 dB).**

**Amburn, S. A., and P. L. Wolf, 1997: VIL Density as a hail indicator. *Wea. Forecasting*, 12, 473–478, doi:10.1175/1520-0434(1997)012<0473:VDAAHI>2.0.CO;2.**

**Greene, D. R., and R. A. Clark, 1972: Vertically Integrated Liquid Water – A new analysis tool. *Mon. Wea. Rev.*, 100, 548–552, doi:10.1175/1520-0493(1972)100<0548:VILWNA>2.3.CO;2.**

**National Weather Service, Vertically Integrated Liquid Density as an indicator of hail size. National Weather Service Nashville Web Page, NOAA/National Weather Service, https://www.weather.gov/ohx/vildensityhailsize.**

**Oue, M., S. M. Saleeby, P. J. Marinescu, P. Kollias, and S. C. van den Heever, 2022: Optimizing radar scan strategies for tracking isolated deep convection using observing system simulation experiments. Atmos. Meas. Tech., 15, 4931–4950, doi:10.5194/amt-15-4931-2022.**

106-114: This mostly describes the input to the cell tracking rather than how the algorithm actually works and any parameter choices. "The MCIT algorithm ingests time series of volume scans and tracks local maxima of VIL by identifying the two cells in consecutive radar scans that have common maximum VIL." This is a rather sparse description of what appears to be quite a sophisticated algorithm as described in Hu et al. (2019a). The authors should describe basic components of any tracking algorithm, such as the minimum size of a cell, the consideration of displacement between consecutive images, and how merging and splitting is handled (whether by the algorithm or by later analysis, L126, noting how many cells are removed).

**We have added a more detailed description of how the MCIT algorithm works on lines126-133.**

112: "isolate" – do you mean "identify"?

**Yes, this has been changed.**

117: Best to refer to this as ETH (echo top height) rather than CTH. What if the highest gate with detectable signal is also the highest elevation? What is the uncertainty on CTH with range? Knowing the VCP would help the reader understand.

**CTH has been changed to ETH. As you state, at distances increasingly close to the radar, the echo top height deviates substantially from the actual cloud top height. If the highest gate with detectable signal is also the highest elevation, one of two things happens: 1) if the location is within 10 km of KHGX and the maximum ETH is the maximum elevation, we provide a set value of 10 km ETH or 2) beyond this distance, we assume that ETH is approximately equal to the actual CTH. The uncertainty is greatest within the 10 km closest to the radar because of the cone of silence (as referenced by you in your first major comment). However, we remove any cells that come within 15 km of KHGX which should help mitigate the effects of the cone of silence. ETH also becomes increasingly uncertain with increasing range and from one VCP to another. Since we do not account for changing VCPs, we are unable to quantify ETH uncertainty based on VCP. We have added discussion of this at lines 330-334.**

118: Similar to CTH (ETH): What if the lowest gate is also in the lowest elevation? What is the uncertainty on Hcell with range?

**The primary driver of uncertain for $H_{cell}$ is the ETH. The WSR-88D network always operates with a minimum elevation angle of 0.5º ± 0.2º. With increasing distance from the radar, the bottom most value will become increasingly elevated, with a maximum altitude above radar level at the edge of our domain of approximately 2 km.**

120: What is the uncertainty in CRatio with range?

**As with $H_{cell}$, the primary cause of uncertainty comes from ETH.**

121: What is the uncertainty in area with range?

**The uncertainty in area with range is dependent on beam spreading, since it is based on the contiguous area of a cell where VIL ≥ -20 dB. At a maximum, the largest distance between radar gates used to calculate these areas is slightly larger than 1 km at a range of 125 km.**

123: Table 1. It seems that just using CTH (ETH) already separates the three classes. Are the other criteria really necessary? What would happen if a cell with CTH (ETH) between 8 and 12 km had a lifetime minimum BT warmer than 250 K (if possible) or a lifetime maximum VIL less than 0 dB (if possible) or a lifetime max CRatio less than 0.75 (if possible)? Is the cell then excluded from the statistics? If so, what number of cells is excluded?

**In any instance where even one variable fails to meet its threshold, it is excluded from analysis. Initially, before applying any thresholding, we have 1,664,215 tracked objects for the analysis period. Applying the thresholding outlined in the paper reduces this number to 48,800 cells (35,995 shallow, 7,935 modest deep, and 4,869 vigorous deep convective cells.) So, the vast majority of tracked objects are rejected. Using ETH as an example; if we were to only use the thresholds outlined, the shallow convective cells may not be substantially affected, but modest and vigorous deep convective datasets would end up with thinner high cirrus clouds or mid-level stratus decks for example. While this strict thresholding may restrict our datasets more than necessary, we feel that it is appropriate given that, even under these strict thresholds, we still have substantial numbers of cases for each case type.**

164: "accelerate with time" – this is not shown as such in Figure 3. Sure, the fastest motions are found later in the life cycle, but what proportion of cells accelerate? You would need to provide the time of maximum motion for each individual cell (and perhaps check that it is significantly greater than the mean or minimum motion) and analyse the spread.

**This section of text has been removed as the additional analysis proved to be too troublesome to complete during the review period.**

160-174: "from the south to east" and "from southwest to east" and "the sea breeze along the Gulf Coast plays a part in storm initiation and propagation" – how is direction defined? Is this southward or southerly, which are opposite directions. Please be precise. It would be helpful to refer to the geography (e.g. the map in Fig.5) to indicate if this direction is typically away from or towards the coast (or even parallel) and hint at the typical direction of the sea breeze.

**We have reworded this section to improve clarity. It is now at lines 177-191**

170: "without further analysis" – out of interest, does MCIT not keep track of the number of VIL minima per cell? That could provide some indication (e.g. a PDF of number of VIL minima per cell).

**The version of MCIT used for this study does not track the number of VIL minima per given cell, but these values can be recorded by modifying the algorithm output and redoing the analysis. However, because of time constraints, revising the MCIT to include this variable was not possible during the manuscript review period.**

182-191: Figure 5 does not seem essential to the paper, with the interesting results shown in Figure 6. As the paper has quite a large number of figures, please consider removing figure 5 and merging the relevant discussion with that for figure 6 (although most relevant points are repeated already).

**Done.**

184: "Houston" – to those unfamiliar with the geography, this is difficult to place (as presumably the centre of the domain coincides with the radar location, which may not be in the centre of Houston. Could the authors consider adding at least a marker if not an outline of the urban area to those figures that show the map? Similarly, "Galveston Bay" is perhaps a bit more obvious given the shape of the coastline, but would still be helpful to indicate (or describe its location on the map).

**We have added a red star to all maps to denote the location of downtown Houston and enlarged the black star which shows the location of KHGX. We also added a snippet of text to clarify the location of Galveston Bay.**

188-189: "local maximum over the Houston metropolitan area" – to what extent could aerosol loading be differentiated from an urban heat island effect? Should urban heat island not be mentioned here as a possible cause?

**It is mentioned as a possible cause. The following sentence states that the maximum in initiation "…could be caused by the enhancement from aerosol loading and/or urban heating, but will need further examination in the future." This is found at lines 207-208**

212: "below 5 km, suggesting warm precipitation processes" – Here, or earlier, please provide the average height of the freezing level (and perhaps -20 deg C) during the season.

**HRRR NSE analyses has been removed and the closest National Weather Service office that launches regular soundings is approximately 200 km to the east-northeast of KHGX. As such, this statement has been removed.**

218-219: "The rapid change in GOESBT and HdBZmax shows the quick vertical evolution of these cells resulting in cold precipitation processes." – Again, that's not precisely what is shown. A change in a population of cells does not show a quick change in individual cells. What is shown, instead, is that there is a shift in the population dominated by newly initiated cells (as indicated by Figures 4 and 6) to a population dominated by mature and/or long-lived cells. This is still a very interesting result. The "quick vertical evolution" is still a valid conclusion, but it is more evident from the next Figures and should therefore be mentioned in section 3.4.

**This change has been made. It can be found at lines 262-263.**

239-243: It is not clear why the discontinuity is "unnatural" and not just an indicator of a part of the life cycle. Wouldn't lower Hdbzmax (whether or not due to bright band) simply be an indicator of the storm evolving into an "orphan anvil" that is precipitating? It's not clear why this is presented as an artefact of the analysis, rather than a useful result about the storm physics.

**The discontinuity that we are specifically referring to is most obvious in figure 8b,e. Since these features appear early in cell lifetimes, the likelihood of orphan anvils being present, especially at a height of around 6 km, seems unlikely. That said, we have added a caveat that this discontinuity may also be part of storm physics. It can be found at lines 257-258.**

249-251: "the feature of low dBZavg and high dBZmax is indicative of anvil generation" – How? The use of dBZ-avg for studying anvil generation requires justification. No doubt it is sensible, but to the reader interested in anvil but not so well versed in radar meteorology, this will be very confusing. Is there a reason why the authors did not use the cell area?

**We added text to justify why dBZavg and dBZmax were used at lines 268-272. Comparing two different values for dBZ was chosen over comparing to area to see how far the mean deviated from the one-to-one line depicted in this figure. This analysis technique also allowed us to identify the point in cell normalized lifetimes when convective core dissipation occurred, which would not be possible when comparing with the storm area. We have also included the plots of area versus dBZmax and area vs dBZavg for all three storm types below.**

**Shallow**

[Figure]

[Figure]

**Modest Deep**

[Figure]

[Figure]

**Vigorous Deep**

[Figure]

**dBZmax versus area are shown on the left and dBZavg versus area are shown on the right.**

263-273: Given that the interest is in the patterns during the life cycle, Figure 11 and 12 are not essential to the paper and could be removed. Figure 13 is of some interest, but may appear as supplementary information as it explains an oddity, but is not essential to the main results. Removing this paragraph also presents a more natural flow into the discussion of Figure 14.

**We have removed figures 11, 12, and 13. The reason for removing figure 13 and its discussion is because, while it may be of some interest, it is not critical to understanding the results of these papers, as it is not applied outside of these analyses. The remaining figures have been renumbered and the text adjusted for a natural flow into what was figure 14.**

304: "within these cells" – which cells? Shallow, deep, all?

**This has been addressed to improve clarity. Line 315.**

304-318: This is an interesting result but requires some further serious consideration, particularly in light of discussing uncertainty in height estimates (major comment 2). It is a stretch to suggest clear maxima in ascent rates when the mode is obviously (close to) zero throughout.

**We have addressed this and added discussion about the uncertainty in these results based on ETH and $H_{dBZmax}$ uncertainty. It can be found at lines 330-334.**

Figures 3, 5-15, 18: Please change these to a colour scale that is appropriate for sequential increases and appropriate for colourblind people. Certain colours such as the dark purple, dark green, and black stand out because they contrast with the adjust colours, likely emphasising the wrong data. Most software packages should suggest such colour bars, or one can be designed using a tool such as colorbrewer.

**Done.**

Editorial comments:

17: "the maximum radar reflectivity and its height" – "the height and value of the maximum radar reflectivity"

**Done.**

48: "The land-sea breeze circulation […] have been shown" – "…has been shown"

**Done.**

54: "Previous studies […] suggests" – "suggest"

**Done.**

80: "The area used for this study was selected such that it was centered…" – "Our study domain was centered…"

**Done.**

102: "using the (1)" – "using equation (1)"

**Done.**

111 & 116: Please refer to KHGX rather than WSR here.

**Done.**

126: "ration" – "ratio"

**Done.**

156-157: "All three case types…" – more or less repeats L155, suggest merging.

**Done.**

241: ARL – this acronym is hardly used so please just spell out "above radar level" for the remaining cases.

**Done.**

**Thank you so much for your thorough review of this paper and the suggestions you have made to improve the readability, clarity, and quality of the research.**

The study describes basic cloud properties based on the MCIT cell tracking algorithm. This is useful and timely research, but it could benefit from the following comments:

1. The authors should acknowledge that the core algorithm of MCIT is based on Rosenfeld (1987), and an associated very similar study of the relationships between the tracked cloud properties based on it (Gagin et al., 1985).

    **Done.**

2. Line 24-37, another key reason models usually fail in a real case simulation is that modelers mostly focus on reflectivity comparison between model and radars. Reflectivity can be a very confusing parameter for cloud microphysical analysis since it is proportional to the first moment of particle concentration and $6^{th}$ moment to the size of particles within each radar gate volume. Recent efforts to use forward operators (Ryzhkov et al., 2011, Wolfensberger and Berne., 2018, and Kumjian et al., 2019) to simulate dual polarimetric parameters like ZDR, KDP, and Rhohv demonstrate stronger confidence in cloud microphysical analysis.

    **Added this discussion to the introduction at lines 29-33 in the new version of the manuscript.**

3. Line 62-65, Hu et al., 2019b indicate the dataset is roughly 3000 cells during a multi-year window of cell tracking within the greater Houston area. This study did analyze general characteristics (Figs 2,3, 6-8) of cloud lifecycles of many cells. So I won't say this is only "a few convective clouds".

    **We reworded this section to address your comment. It can be found at lines 65-67.**

4. Line 68, add space "aroundthe".

    **Done.**

5. Line 80-82, it is recommended to use radar site centric domain instead of city landmarks to avoid radar beam size inhomogeneity at the same distance from the center of domain. In addition, the authors domain is over 100 km from the radar. Please justify the vertical extent the authors are focusing on and why.

    **Apologies, the domain is radar centric (the domain center is KHGX). This section was worded incorrectly and has been fixed to reflect this. As for the vertical extent, we are investigating the depth of the troposphere. Cells within 15 km of the radar have been removed based on major comment 1 from reviewer #2. We are focused on**

**investigating the vertical evolution of shallow, modest deep, and vigorous deep convective storms through their entire depth in order to better understand how they evolve over time in a bulk sense.**

6. Line 92-93, what is the vertical resolution for VIL calculation?

   **It varies with range from the radar. The lowest resolution for VIL is at the edge of the domain and conversely highest near the radar. However, because of the cone of silence, VIL calculations within several kilometers of KHGX are not valid because they do not consider the entire depth of a given cell at those distances.**

7. Equation 1, Is there any Z limit for VIL calculation? Say if Z is 60 dBZ, is it still used to calculate VIL?

   **No, there is no limit for the VIL calculation.**

8. Line 117, for CTH detection, what reflectivity threshold, if any, is used here? Or how did the authors determine if the 88D radar data is noise or weather echoes?

   **The reflectivity threshold for CTH (changed to ETH based on comments from reviewer #2) is -10 dBZ. This has been added to the manuscript to improve clarity at line 138.**

9. Line 118, the lowest gate of radar detectible signal is forced by the range from radar as well, so how did the authors make a correction about that?

   **Corrections were not made and values for H$_{cell}$ are affected by this issue. The maximum height of a gate at the edge of our domain (ignoring the effects from any ducting), is approximately 2 km above radar level since KHGX scans at the lowest elevation of 0.5°.**

10. Table 1, is it possible for the authors to provide a time series or movie of one of the tracked cases to demonstrate the labeling of shallow, modest, and vigorous convective cells? Please include the cell boundaries as contours when generating the figure/movie, in the supplementary materials would be sufficient.

    **We have included animations of VIL (dB) and the tracked cells from a day where shallow, modest deep, and vigorous deep cells were prevalent. On this day, we analyzed 258 shallow, 71 modest deep, and 81 vigorous deep convective cells.**

KHGX - 2020/07/22 10:36:48

11.    Figure 4. Are there any restrictions on the lifetime of cells here? Say at least 30 min? Or 5-6 radar volume scans? In addition, please elaborate on the reasoning for peak differences between the three convection types.

**No, there are no restrictions on cell lifetime. All of the restrictions placed on cells are all outlined in table 1. Below we have included the figure 4 with and without a 30-minute lifetime threshold for each case type. While there is a substantial reduction in the number of cases (especially for shallow convective cells, the distributions of initiation times remain relatively unchanged.**

**Figure 4 (no lifetime filter)**

[Figure]

**Figure 4 (30-minute lifetime threshold)**

**12.** Figure 8, for the shallow case, why there's not much difference at different life stages? A similar concern applies to the vigorous type of convection. For example, in panel i, the majority of cell max Z is still very high over 8 km (from 10-over 50 dBZ) here. This raises concerns about the cell tracking quality of this dataset. Cell dissipating should not have Z still over 50 dBZ, that is a hail signature, but in panel l, Z over 50 dBZ is over 10 km and has the highest frequency, although it is less frequent compared with panel i, but still the highest under this normalized lifetime category. It seems that the tracking was terminated due to a splitting event or continuing under another cell identity. This problem was addressed and fixed by Yin et al. (2022). This is an improved version of MCIT that keeps track of all the splits.

**We are not sure why there is not much difference at different life stages for shallow convection. We have included a figure below that recreates figure 8, but for 10-minute bins for the first 30 minutes of cell lifetimes rather than 0.25 normalized lifetime bins. This supplementary figure shows the same general trends as figure 8 and captures 89.4% (127,906 of 142,923 volume scans) of all shallow cases, 37.6% (34,917 of 92,798 volume scans) of all modest deep cases, and 23.2% (22,075 of 95,219 volume scans) for all vigorous deep cases.**

[Figure]

There is a shift to the upper right for subplots d and g in figure 8 and the above figure, which starts and returns to the bottom left early and late in shallow cell lifetimes, respectively. As far as vigorous convection goes, while there are obvious shifts over the course of the lifetimes of these cells in figure 8, you are correct in that there are anomalous values still present. We have attempted to mitigate this by removing cells which begin or end their life along a domain boundary, but this does not seem to have completely addressed the problem. Some of the modifications we made to the MCIT algorithms were also with respect to handling splits and merges. The specifics of the changes we made to the MCIT algorithm can be found below.

In cell identification, an ambiguous situation arises when iterating over all cells when more than one cell (cell A and cell C) considers the same neighbor (cell B) as a candidate for cell merging. In this case, the algorithm has been modified to merge the cells with the same neighbor (cells A and C merge with cell B) and with one another (cells A and C merge together), recursively. In cell tracking, an ambiguous situation may arise in two different scenarios:

**A cell from map(t+1) is the potential split or continuation of two or more cells from map(t). In this case, continuation has been set to prevail over split situations. Nevertheless, if different cells from map(t) are the potential source of the cell from map(t+1), the cell with maximum integrated common VIL is the one selected.**

**A cell in map(t) has two or more potential split or continuity cells in map(t+1). In this case, as before, continuation prevails over split situations, but if different cells are candidates, the cell with maximum integrated common VIL is the one selected. Only one cell can be defined as continuity, the rest are labelled as splits.**

References:

Ryzhkov, A., M. Pinsky, A. Pokrovsky, and A. Khain, 2011: Polarimetric Radar Observation Operator for a Cloud Model with Spectral Microphysics. J. Appl. Meteor. Climatol., 50, 873–894, https://doi.org/10.1175/2010JAMC2363.1.

Wolfensberger, D. and Berne, A.: From model to radar variables: a new forward polarimetric radar operator for COSMO, Atmos. Meas. Tech., 11, 3883–3916, https://doi.org/10.5194/amt-11-3883-2018, 2018.

Kumjian, M. R., C. P. Martinkus, O. P. Prat, S. Collis, M. van Lier-Walqui, and H. C. Morrison, 2019: A Moment-Based Polarimetric Radar Forward Operator for Rain Microphysics. J. Appl. Meteor. Climatol., 58, 113–130, https://doi.org/10.1175/JAMC-D-18-0121.1.

Rosenfeld, D., 1987: Objective method for analysis and tracking of convective cells as seen by radar. Journal of Atmospheric and Oceanographic Technology, 4, 422-434.

Gagin, A., D. Rosenfeld and R.E. Lopez, 1985: The relationship between height and precipitation characteristics of summertime convective cells in south Florida. Journal of Atmospheric Sciences, 42, 84-94.

Yin, J., Pan, Z., Rosenfeld, D., Mao, F., Zang, L., Zhu, Y., Hu, J., Chen, J. and Gong, J., 2022: Full-tracking Algorithm for Convective Thunderstorm System from Initiation to Complete Dissipation. *Journal of Geophysical Research: Atmospheres*, p.e2022JD037601.

**Thank you for your comments on this manuscript that have helped to clarify and improve the quality of the analyses presented therein.**

---

## Author Response (AR2)

**Suggestions for revision or reasons for rejection**

I commend the authors for their efforts responding to the reviewers' comments. These comments have been dealt with in an appropriate manner and have provided sufficient clarification (also in the manuscript) that my recommendation is to accept this for final publication.

I will suggest a set of (related) technical corrections:

1. Report ascent rate in m/s rather than km/min, as the former is more immediately recognisable. This affects Figure 11 and line 402.

**This suggestion has been completed and is reflected in the most recent revision of the paper.**

2. Regarding Figure 11, I do not see the "dashed line" mentioned in the caption.

**The dashed line has been made thicker to be more apparent on the plots in Figure 11.**

3. To make the ascent rate discussion slightly more statistically interesting/robust. The authors could consider including a set of quantiles (e.g. 25th, median, and 75th) in Figure 11, to better understand the likelihood of large ascent rates or descent at different stages of the life cycle.

**We have added a median line (solid black) and 10th and 90th percentiles lines (dotted black) to Figure 11 to further elucidate the likelihood of larger ascent/descent rates from the methodology presented.**

**Thank you for your additional feedback to improve the quality and readability of this paper.**

Suggestions for revision or reasons for rejection

All line numbers refer to the tracked version of the manuscript.

Major comments:

The authors attempt to show climatology statistics of convective cells near the Houston, TX area using 4 years of radar and satellite data during the summer seasons. A modified MCIT tracking algorithm was used. My main concerns are as follows:

(1) Little radar data quality control was performed, such as attenuation correction and signal-noise-ratio (SNR) within the polar coordinates before regridding the data into Cartesian coordinate. In addition, key variable VIL should limit Z to 56 dBZ as introduced by Greene and Clark (1972), this will also require a rerun for the entire dataset and tracking.

**We have re-run the dataset using a $Z_H$ limit of 56 dBZ and remade all of the figures. In short, there was essentially no change to the results from adding this $Z_H$ limit. As for attenuation corrections, since KHGX operates at S-Band, as do all WSR-88D radars, the effects of attenuation are minimal and would only be present with the most extreme convection. As for SNR, the data freely provided from the WSR-88D network does not contain SNR and these corrections cannot be done.**

(2) Using AOD as proxy for aerosol condition within clouds is a poor choice. AOD is a column integrated optical product, its own bias as to cloud invigoration cannot be quantified. According to Stier (2016) AOD explained only 25% of the CCN variance, not to mention the underlying key microphysical parameters like droplet concentration. It is suggested to use Rosenfeld et al., 2016 cloud base retrieved CCN concentration as a more direct aerosol signal to cloud invigoration statistics.

**The method suggested in Rosenfeld et al. (2016) uses polar orbiting satellites, which would severely limit the analyses presented herein. Using GOES-16 AOD including only "medium" and "high" quality measurements for calculating a 30-minute pre-cell initiation mean AOD value for the location of cell initiation already reduced our shallow convective dataset by 93.4% (from 35,974 to 2,361 cells), reduced our modest deep convective dataset by 96.2% (from 7,930 to 303 cells), and reduced our vigorous deep convective dataset by 95.6% (from 4,869 to 212 cells). Using polar orbiting satellite data would limit our sample size even further and may leave us with only a handful of cases or none at all. These reasons are why we chose to go with GOES-16 AOD as a proxy for aerosol concentration.**

(3) The methodology of dividing the cells into shallow, modest deep, and vigorous deep require further justification or maybe simplification here. See detailed comments below.

**The reasoning for dividing into three categories rather than just "shallow" and "deep" is because deep convection can vary substantially in intensity. As shown by the analyses presented herein, simply dividing deep convective cases into these two groups shows noticeable differences in cell characteristics. The echo top height is used as the discriminator because the other thresholds were originally designed to do as you suggest and simply divide into shallow and deep convection, but adding the different echo top height thresholds to further divide into "modest" and "vigorous" deep convective cells showed clear differences in behavior that we believe is important in discriminating between weaker and stronger deep convection.**

(4) The fonts for most figures are too small to read. Please modify.

**Fonts were enlarged on most figures to improve readability.**

As my main concerns will require a rerun of the overall dataset, I will stop here and continue review process once the updated dataset is ready.

**The concerns mentioned in your major comments have also been further addressed below while responding to your minor comments.**

Minor comments:
1. Line 10, "Radars have been traditionally used to provide the convective clouds characteristics." This statement is a bit odd here. Radar was first invented during war time and was used for missile detection instead of weather. In addition, once radar is applied in meteorology studies, both warm and cold season weather phenomenon are studied without priority rankings.

**Reworded this sentence to provide clarity of meaning.**

2. Line 14, consider changing "warm" to "summer" as Houston is warm from April to almost the end of Nov.

**We used "warm" here since summer is typically considered June, July, and August, whereas September begins meteorological autumn. Therefore, we have chosen to leave the wording as is.**

3. Line 36-42, it is a bit confusing here what modelers are really missing during the debate of warm and cold phase convective invigoration, can you elaborate on what is being debated here? In addition, doesn't the models intercomparison project from van den Heever's group show the models cannot agree on each other in terms of precipitation around Houston on the same set up and case? This sounds like the modelers are more debating on their model's inconsistence instead of an invigoration theory.

**We have added information to further describe warm- and cold-phase invigoration and inconsistency among models. While we agree with your point, this study is not meant to investigate model inconsistencies, rather it is meant to provide climatological analyses of convective case types for easy comparison with model output.**

4. Line 49 – 50, TRACER and ESCAPE were choosing Houston mainly for the variation of aerosol conditions here. In addition, it is well equipped with WSR-88D radar coverage, LMA coverage, TCEQ network, etc. Basically, we are treating Houston as a natural laboratory to study aerosol induced microphysical processes here. Please modify and include this in the text.

**We have added text to include the importance of the variation in aerosol conditions.**

5. Line 91-95, Houston is a natural lab as stated earlier due to its variability in terms of aerosol conditions. Although the authors identified the local pollution source, it is not reasonable to simply state SW is pristine, and NE is polluted. The state of pollution is not only determined by local sources but also synoptic weather conditions. In other words, according to Rosenfeld et al, 2016's satellite retrieval technique, both SW and NE of Houston can be polluted or pristine in a case-by-case scenario.

**While we agree with the sentiment here, going through and manually identifying the surface flow for all hours analyzed is not reasonable. The reasoning for simplifying as we have is based on the general flow across the Houston area during the months considered. We have added text that specifies that increasingly large errors may come about as the flow in a given case deviates more from the general flow.**

6. Line 99, using AOD as a proxy for aerosol within cloud or related to cloud invigoration can be quite noisy and subject to false conclusions. AOD is a column integrated optical product, its own bias as to cloud invigoration cannot be quantified. In an extreme case, one can expect a high AOD case while the aerosol contributed to is over/under the cloud column entirely. In other words, AOD has little to do with the CCN actually got activated into cloud droplets.

**To combat the noisy nature of AOD, we only used AOD data that were denoted as "medium" or "high" quality observations. We then collected these values for the 30 minutes prior to cell initiation and used the mean AOD value as the cell's AOD value for initiation. We never try to separate CCN specifically from the total aerosol population. AOD are also not collected when clouds are present which mitigates the concern of AOD being over/under clouds. We were aiming to maintain the largest number of cases possible by using this method. Using the method presented in Rosenfeld et al. (2016) with a polar orbiting satellite would substantially reduce the number of cases analyzed here and would introduce huge temporal discrepancies between the time of satellite flyover and cell initiation.**

7. Line 113, GOES BT13 is an IR product and has a resolution of 2 km and has a 5 min resolution. So what procedure has been done to match the radar and satellite data both in space and time? Simply the nearest neighbor perhaps? Then how much bias does this procedure will introduce to the overall dataset?

**The GOES data were regridded to the same grid as ZH from KHGX. We used nearest neighbor temporally. While this may introduce some error, since the scan time of KHGX and GOES-16 are approximately the same, the longest possible difference between these two datasets is 2.5 minutes. We have added text to mention this.**

8. Line 120-125, For VIL calculation, Z should be capped at 56 dBZ, as introduced by Greene and Clark 1972 to avoid possible ice phase hydrometeor contamination. Please rerun the cases with this threshold.

**We have rerun the cases with the capped threshold and the results have not changed. Text has been added to mention the 56 dBZ cap.**

9. Line 147, ETH definition should not be a fixed Z>-10 dBZ as the SNR degrades the signal with distance. Please use an SNR threshold instead here, say SNR > 10 or 15 dB. This step should be done in the polar coordinate before regrid the dataset into Cartesian coordinate.

**Level-2 WSR-88D data does not contain SNR. Therefore, we cannot complete this request.**

10. Line 149-150, according to the authors description, Hcell is not corrected with increasing range, then if the authors use cells 100 km from the KHGX, the base scan tilt is already approaching melting layer height (4 km), this is quite strong simplification suggested in the manuscript and subject to underestimation of Hcell. In addition, Hcell depth should not use detectible signal without quality control, but like ETH, use SNR masked signal.

**We agree that $H_{cell}$ is subject to underestimation with increasing distance. However, this underestimation would affect distant shallow cells primarily, not deep convective cells. Given the number of shallow cells we are analyzing herein, losing some shallow cells toward the edges of our domain is acceptable. As mentioned previously, level-2 WSR-88D data does not contain SNR, and as such, we cannot apply an SNR mask.**

11. Table 1., it seems the only difference between Modest Deep vs Vigorous Deep is the lifetime max ETH difference. Why is it? What's the author's justification and objectives here? Would it be simpler if you combine the two deep scenarios? In addition, why does shallow convection cells are limited to 30 km2? Any justification? Sounds a bit random.

The empirically-derived thresholds in Table 1 are there only to ensure that we are looking at shallow or deep cells. The reason for the difference in ETH only for deep convection is as follows: the cell has already reached "deep convection" status if all other thresholds are satisfied. However, not all deep convection is created equally. Modest deep convective cells are meant to capture convective cells which have reached a deep convective state, but are not intense enough to grow into the most intense convective storms that reach the tropopause. As shown in our analysis, while modest deep convective cells share many similarities with vigorous deep convective cells, there are also stark differences between them. Shallow cells are limited to an area of 30 km2 to prevent tracking large shields of stratiform precipitation, which shares many of the existing thresholds we identified to isolate shallow cells. The area threshold for shallow cells ensures that we are looking at a small, discrete cell, not a large cohesive blob of stratiform rain.

Thank you for your in-depth comments to improve the quality of this manuscript.

---

## Author Response (AR3)

**Public justification (visible to the public if the article is accepted and published)**:
The abstract needs a minor revision to summarize your findings concerning the influences of aerosol on the different types of clouds as discussed in the main body and summary.

**We have added discussion about aerosols and changed some wording to improve clarity. Thank you for your suggestion to help improve this manuscript.**